# Evidence for additive and synergistic action of mammalian enhancers during cell fate determination

**Jinmi Choi[1], Kseniia Lysakovskaia[1], Gregoire Stik[2], Carina Demel[1], Johannes Söding[3], Tian V Tian[2], Thomas Graf[2], Patrick Cramer[1]\***

[1]Max Planck Institute for Biophysical Chemistry, Department of Molecular Biology, Göttingen, Germany; [2]Gene Regulation, Stem Cells and Cancer Program, Centre for Genomic Regulation (CRG), Barcelona, Spain; [3]Max Planck Institute for Biophysical Chemistry, Quantitative Biology and Bioinformatics, Göttingen, Germany

**Abstract** Enhancer activity drives cell differentiation and cell fate determination, but it remains unclear how enhancers cooperate during these processes. Here we investigate enhancer cooperation during transdifferentiation of human leukemia B-cells to macrophages. Putative enhancers are established by binding of the pioneer factor C/EBP$\alpha$ followed by chromatin opening and enhancer RNA (eRNA) synthesis from H3K4-monomethylated regions. Using eRNA synthesis as a proxy for enhancer activity, we find that most putative enhancers cooperate in an additive way to regulate transcription of assigned target genes. However, transcription from 136 target genes depends exponentially on the summed activity of its putative paired enhancers, indicating that these enhancers cooperate synergistically. The target genes are cell type-specific, suggesting that enhancer synergy can contribute to cell fate determination. Enhancer synergy appears to depend on cell type-specific transcription factors, and such interacting enhancers are not predicted from occupancy or accessibility data that are used to detect superenhancers.

**\*For correspondence:**
patrick.cramer@mpibpc.mpg.de

**Competing interests:** The authors declare that no competing interests exist.

## Introduction

Enhancers are cis-regulatory DNA elements that drive the transcription activity of target gene promoters (*Beagrie and Pombo, 2016*; *Long et al., 2016*; *Spitz and Furlong, 2012*). Enhancers contain transcription factor (TF) binding sites, recruit TFs, and drive cell type-specific gene expression programs. Previous studies have defined lineage-determining clusters of enhancers ('superenhancers') that span several kilobases of DNA and contain a high density of TF- and mediator-binding sites (*Hnisz et al., 2013*; *Whyte et al., 2013*). It has been suggested that the individual constituent enhancers of such clusters cooperate in a synergistic manner to activate target genes (*Hnisz et al., 2017*; *Shin et al., 2016*). Cooperation may be achieved by liquid–liquid phase separation of general and gene-specific TFs (*Hnisz et al., 2017*).

Enhancer cooperation has been investigated at the level of individual genes. Available studies suggest that enhancers can cooperate in an additive or a synergistic manner. Genetic in vivo dissection of the enhancer elements of the $\alpha$-globin superenhancer has shown that the activity of the $\alpha$-globin genes increased linearly with the number of enhancers used, showing additive enhancer cooperation (*Hay et al., 2016*). Additive cooperation was also demonstrated for enhancers within the Myc-regulating superenhancer and for limb enhancers (*Bahr et al., 2018*; *Osterwalder et al., 2018*). Non-synergistic enhancer cooperation was further observed in mouse embryonic stem cells (*Moorthy et al., 2017*) where deletion of an individual enhancer resulted in only a small effect on target gene expression (*Moorthy et al., 2017*). There is also evidence that two enhancers can synergistically activate selected target genes (*Fulton and van Ness, 1994*; *Guerrero et al., 2010*;

*Maekawa et al., 1989*; *Stine et al., 2011*). Moreover, multiple enhancers can interact simultaneously with their target gene promoter in mouse and human cells (*Allahyar et al., 2018*; *Oudelaar et al., 2018*). Evidence for enhancer–enhancer interactions was also obtained in *Drosophila* (*Lim et al., 2018*).

Despite these studies, the functional cooperation between enhancers over time has not yet been studied in a native genomic context and a genome-wide manner. As a consequence, it is unknown to what extent enhancers cooperate dynamically in cells and whether they do so additively or synergistically or both. To study this, enhancer and promoter activity must be monitored over time with a non-perturbing genome-wide method. We have previously reported such a method called transient transcriptome sequencing (TT-seq). TT-seq combines short-term metabolic RNA labeling (5 min) with sequencing of newly synthesized RNA fragments and provides a genome-wide unbiased view of RNA synthesis activity (*Schwalb et al., 2016*). The fragments are derived from all RNA species, including short-lived non-coding RNAs such as enhancer RNA (eRNA) and messenger RNA (mRNA) (*Schwalb et al., 2016*).

TT-seq can monitor changes in enhancer and promoter activities over time with great sensitivity. During T-cell stimulation, transcription from enhancers and promoters of responsive genes is activated simultaneously (*Michel et al., 2017*). Enhancers can be paired with their putative target gene promoters based on their proximity (*Michel et al., 2017*). Using eRNA production as a proxy for enhancer transactivation activity (*Henriques et al., 2018*; *Mikhaylichenko et al., 2018*), TT-seq is very well suitable to identify active enhancers, to pair enhancers with their putative target promoters, and to measure the transcription activity of enhancers and promoters genome-wide. Putative enhancers can be detected by mapping of chromatin signatures (*Creyghton et al., 2010*; *Heintzman et al., 2007*; *Robertson et al., 2008*). However, these techniques have limitations if time-resolved analysis in a dynamic system is required. Also, enhancers can be removed by genome editing but this is not readily possible for thousands of putative enhancer regions.

To address the question of enhancer cooperation during cell type determination, we used a transdifferentiation system driven by a single TF (*Rapino et al., 2013*). In this system, induction of the TF C/EBPα converts human leukemic B cells into macrophage-like cells within 7 days in a nearly synchronous and efficient manner (*Rapino et al., 2013*). Transdifferentiation involves dramatic changes in gene expression, including silencing and activation of cell type-specific genes (*Rapino et al., 2013*). At the end of the process, cells gain the characteristics of macrophages, including acquisition of adherence, phagocytic activity, and inflammatory response (*Gaidt et al., 2016*; *Rapino et al., 2013*). Due to these properties, the C/EBPα-induced transdifferentiation system enables quantitative data acquisition and analysis and is ideally suited for addressing mechanistic questions on enhancer function and cooperation in vivo.

Here we report a genome-wide multi-omics data set for C/EBPα-induced transdifferentiation of human B cells to macrophages . In-depth analysis of the data provided general insights into the order of events that establish active enhancers. Furthermore, TT-seq allowed us to identify transcriptionally active enhancers, to pair these with their putative target promoters, and to analyze the changes in transcription activity of enhancers and promoters over time. Our analyses revealed that most enhancers that drive the expression of a common target gene do so in an additive manner. However, for about one-fifth of the enhancers tested (136 in total), the change in transcription activity of the target gene was larger than the change attributable to the sum of the enhancer activities, indicating that enhancers cooperate synergistically at these loci to drive target gene transcription.

## Results

### RNA-seq reveals two distinct transitions during transdifferentiation

To study the temporal order of gene regulatory events during transdifferentiation of human precursor leukemia B cells to macrophage-like cells, the master TF C/EBPα fused to estrogen receptor were activated by addition of beta estradiol (*Figure 1A*). As revealed by FACS analysis of the macrophage marker CD14 and the B cell marker CD19, as well as additional markers monitored by RT-qPCR, transdifferentiation was efficient and occurred in a nearly synchronous manner (*Figure 1—figure supplement 1A and B*). We then applied several genome-wide techniques (*Figure 1B*) to monitor RNA metabolism during transdifferentiation. RNA-seq at 0, 12, 24, 30, 36, 48, 72, 96, and 168 hr

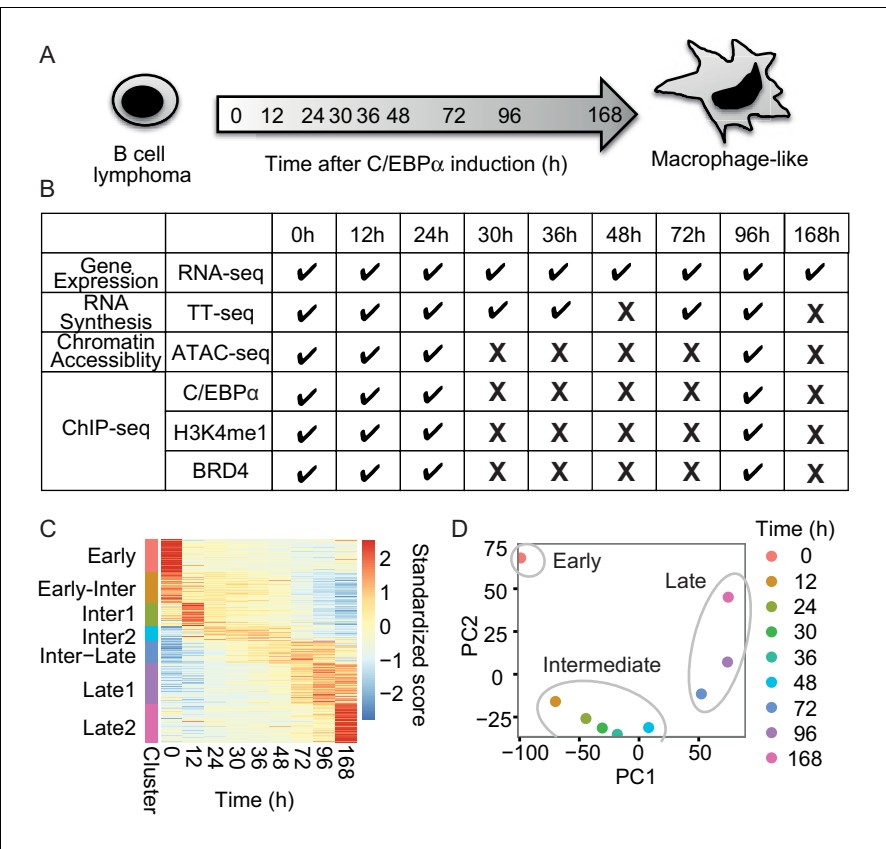

**Figure 1.** RNA-seq monitors two transitions during transdifferentiation. (**A**) C/EBPα induces transdifferentiation from precursor leukemia B-cell to macrophage-like cells. (**B**) Genome-wide data sets collected at different time points during transdifferentiation. (**C**) Heatmap of differentially expressed genes (n = 5516) from RNA-seq (|log2 (FC)|>1, FDR < 0.05) for seven clusters obtained with k-mean clustering. (**D**) PCA plot of differentially expressed genes from RNA-seq during transdifferentiation.

The online version of this article includes the following figure supplement(s) for figure 1:

**Figure supplement 1.** Transcription activity during transdifferentiation.

after induction revealed that a total of 5516 protein-coding mRNAs changed their levels significantly (false discovery rate [FDR] < 0.05) and by at least a twofold change (|log2(FC)|>1) (*Figure 1C*). Upregulated genes were enriched for the terms signaling and immune system processes, whereas downregulated genes were enriched for chromosome organization and cell cycle functions (*Figure 1— figure supplement 1C*), consistent with the fact that macrophages become quiescent (*Rapino et al., 2013*).

Changes in gene expression occurred in two major transitions, as indicated by principal component analysis and Euclidean distance analysis of RNA-seq data (*Figure 1D* and *Figure 1—figure supplement 1D*). The first transition occurred rapidly, between 0 and 12 hr, and was followed by a late transition from 48 to 72 hr (*Figure 1D*). Therefore, the time points of 12 hr and 48 hr represented intermediate states. The variance of the first principal component (60.9%) corresponded mainly to genes involved in immune system processes, whereas the variance of the second component (35.7%) was mainly due to transiently responsive genes involved in developmental processes (*Figure 1—figure supplement 1E and F*). The data also identified two sets of genes exclusively expressed either in B-cells (cluster 'Early', 896 genes) or in macrophage-like cells (cluster 'Late1' and 'Late2', 1083 and 1046 genes), which we refer to as 'cell type-specific genes' (*Figure 1C*). Based on the observed changes in RNA levels, we divided the transdifferentiation process into an early transition, which leads from B-cells to an intermediate state, and a late transition, which leads from an intermediate state to macrophage-like cells. The classification of two distinct transitions facilitated further analyses of cell type-specific gene expression during transdifferentiation.

## TT-seq monitors transcriptionally active putative enhancers

We performed TT-seq at 0 hr, 12 hr, 24 hr, and 96 hr post induction to monitor changes in newly synthesized RNA, including eRNAs (*Figure 1B*). We also measured chromatin accessibility, that is, regions of nucleosome depletion, using the Assay for Transposase-Accessible Chromatin (ATAC-seq). We further used chromatin immunoprecipitation (ChIP-seq) to monitor changes in genome-wide occupancy with C/EBPα, and monomethylation at histone H3 lysine 4 (H3K4me1) as a marker of primed enhancers (*Figure 1B*).

We then used TT-seq data to annotate transcription units (TUs) as described (*Michel et al., 2017*), and identified 43,193 TUs (Materials and methods). Of these, 9993 and 586 had been annotated as mRNAs and lincRNA, respectively, by GENCODE (*Harrow et al., 2012*). Other TUs were classified as downstream (ds) RNAs, upstream antisense (ua) RNAs and convergent (conv) RNAs based on their locations with respect to mRNAs (4033, 1896, and 555 TUs, respectively). Of the remaining 26,130 TUs, 8165 fell into regions that were marked by H3K4me1 and depleted for nucleosomes that do not overlap with more than 20% of transcripts annotated in Gencode (Materials and methods) (*Figure 2—figure supplement 1A*). We therefore refer to their associated RNAs as eRNAs (*Figure 2A*). Regions covered by eRNAs that fell within 1 kb of each other were merged, as exemplified for the −148 kb enhancer of *KLF4* (*Figure 2B*).

The resulting 7624 transcriptionally active regions correspond to putative enhancers (referred to as 'enhancers' for simplicity). These enhancers had a median length of ~1 kb and were predominantly located in intergenic regions (*Figure 2—figure supplement 1B and C*). Among these, 3539 eRNAs significantly changed expression during transdifferentiation ($|log2(FC)|>1$, FDR < 0.05) (*Figure 2—figure supplement 1D*). We grouped enhancers in six clusters with downregulated, transiently produced, and upregulated eRNAs (*Figure 2C*). Among the significantly changed eRNAs,

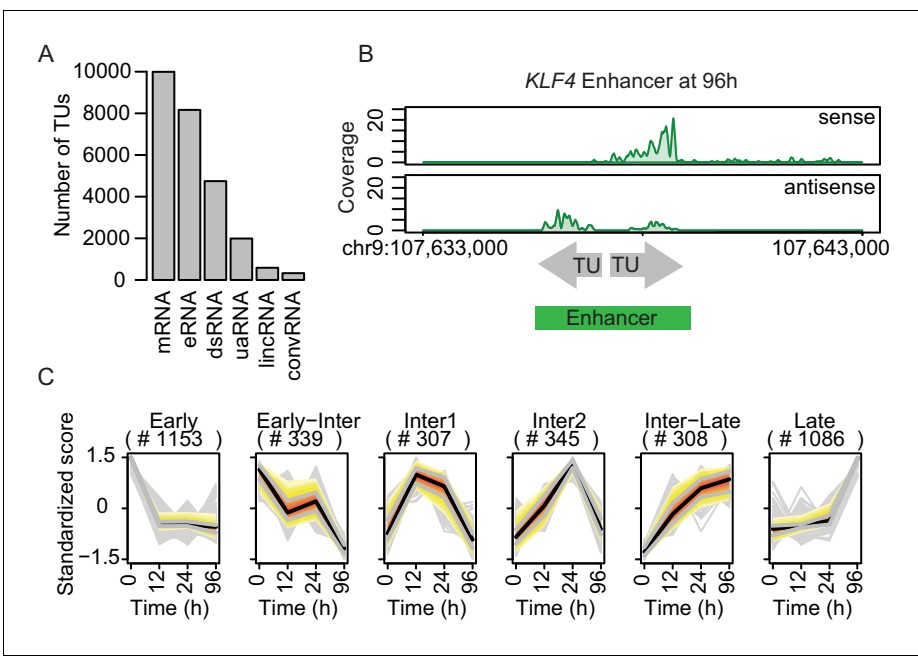

**Figure 2.** TT-seq identifies transcriptionally active enhancers. (**A**) TT-seq identified various classes of RNAs including previously annotated stable messenger RNAs (mRNAs), novel transient enhancer RNAs (eRNAs), and other noncoding RNAs (downstream RNAs (dsRNAs), upstream antisense RNAs (uaRNAs), long intergenic noncoding RNAs (lincRNAs), convergent RNAs (convRNAs)). (**B**) TT-seq coverage tracks on *KLF4* enhancer at 96 hr exemplify how RNAs that were synthesized within 1 kb of another were merged into a single enhancer. (**C**) Dynamically regulated eRNAs ($|log2(FC)|>1$, FDR < 0.05) were clustered using k-means clustering according to the kinetics of synthesis level changes during transdifferentiation. Black line represents median. Red, yellow, and gray represent 0.25, 0.5, and 0.75 quantiles, respectively.

The online version of this article includes the following figure supplement(s) for figure 2:

**Figure supplement 1.** TT-seq identifies transcriptionally active enhancers.

27.1% showed upregulation in the early transition (Cluster 'Inter1', 'Inter2', 'Inter-Late') and 30.7% in the late transition (Cluster 'Late') (*Figure 2C*). Moreover, 32.6% were downregulated in the early transition (Cluster 'Early') while 9.6% were further downregulated in the late transition (Cluster 'Early-Inter' *Figure 2C*). In summary, we used a combination of TT-seq, H3K4me1 ChIP-seq, and ATAC-seq to annotate a conservative set of 7624 transcriptionally active enhancers, of which >46% showed significant changes in eRNA synthesis, with 57.8% being initially upregulated and 42.2% being downregulated (*Figure 2—figure supplement 1D*). As a result, we obtained the dynamics of the enhancer landscape during transdifferentiation.

## C/EBPα binding and chromatin opening

In order to investigate how C/EBPα may induce the observed changes in enhancer landscape and gene transcription, we mapped C/EBPα-binding sites genome-wide by ChIP-seq at 0, 12, 24, and 96 hr. The majority of the obtained 14,561 ChIP-seq peaks (FDR < 0.05) fell into intergenic or intronic regions (*Figure 3—figure supplement 1A*). To investigate the consequences of C/EBPα binding and its impact on enhancers, we studied the changes in H3K4 monomethylation and chromatin accessibility at C/EBPα-binding sites. On 4550 sites with low or undetectable chromatin accessibility at 0 hr, H3K4me1 was present in the absence of C/EBPα binding (*Figure 3A* and *Figure 3—figure supplement 1B*). After 12 hr, C/EBPα was bound at these sites, chromatin became more accessible compared to 0 hr, and H3K4 monomethylation was observed in the regions flanking the site. After 24 hr, chromatin showed increased accessibility at these sites relative to previous time points whereas H3K4me1 decreased, likely due to histone depletion (*Figure 3A* and *Figure 3—figure supplement 1B and C*).

During the transition from 24 hr to 96 hr binding of C/EBPα occurred at a different set of late activated enhancers. These enhancers showed low levels of H3K4me1 and low chromatin accessibility in the beginning of the transdifferentiation process, both of which however increased strongly over time (*Figure 3—figure supplement 1C*). C/EBPα occupancy at 24 hr and 96 hr showed the highest correlation with H3K4me1 at 24 hr and with chromatin accessibility at 96 hr, respectively (*Figure 3—figure supplement 1D*). At these late sites, an increase of H3K4me1 was observed concomitant with C/EBPα binding, but chromatin opening was again delayed as exemplified for the −358 kb *KLF4* and the +349 kb *MAFB* enhancer (*Figure 3—figure supplement 1E*). Together, these results indicate that C/EBPα initially binds to primed enhancers and subsequently acts as a pioneer factor at de novo enhancers, where it increases H3K4 monomethylation and may induce chromatin opening.

## Chromatin opening and eRNA synthesis

Next, we asked whether H3K4 monomethylation or chromatin accessibility is a prerequisite for eRNA synthesis. We clustered 943 significantly upregulated enhancers (log2(FC) >1, FDR < 0.05) occupied by C/EBPα according to changes in eRNA synthesis (*Figure 2C*), and then sorted them within clusters by eRNA signal (*Figure 3B*). We found that eRNA synthesis followed C/EBPα binding, H3K4 monomethylation, and chromatin opening, as seen clearly for the late (96 hr) eRNA cluster (*Figure 3B*). Enhancer transcription always followed chromatin opening. At sites where chromatin was closed at 0 hr and opened after 12 hr, eRNA synthesis was observed only at 24 hr (*Figure 3C and D*). This was corroborated when comparing changes in chromatin accessibility and eRNA synthesis at specific loci. For example, at the −54.8 kb enhancer of *KDM7A*, chromatin opening started 12 hr after induction but eRNA synthesis was only detectable after 96 hr. Similarly, the +37.7 kb *CD14* enhancer showed nucleosome depletion at 24 hr and eRNA production at 96 hr (*Figure 3E*).

To further investigate the order of events, we analyzed the temporal relationship between C/EBPα binding and eRNA synthesis for all enhancers bound by C/EBPα (1487 out of 7624 enhancers). Of these, 965 were differentially transcribed during transdifferentiation compared to 0 hr (log2 (FC) >1, FDR < 0.05, *Figure 3—figure supplement 1F and G*), and 76.2% (735 of 965 enhancers) showed increased eRNA synthesis upon C/EBPα binding. Changes in C/EBPα binding from 12 hr to 24 hr correlated with later changes in eRNA synthesis, from 24 hr to 96 hr (*Figure 3—figure supplement 1H*). At the time of eRNA synthesis, C/EBPα occupancy had decreased relative to the previous time points (*Figure 3B*), suggesting that C/EBPα is required for chromatin opening but not for transcription of late enhancers. In summary, these analyses provided insights into the temporal order of

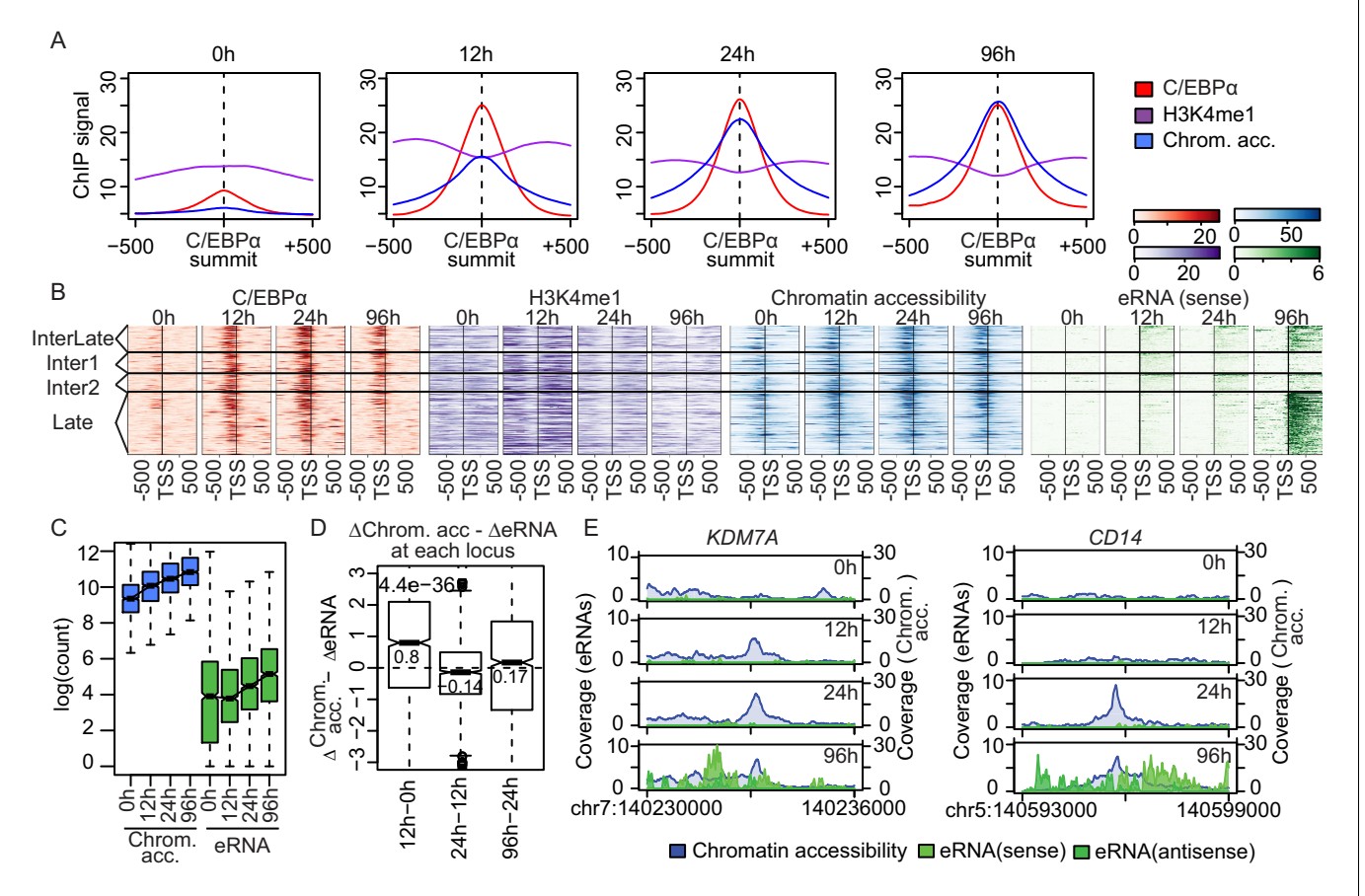

**Figure 3.** Enhancer transcription follows chromatin opening. (**A**) Average signal of ChIP-seq of C/EBPα (red) and H3K4me1 (purple) or ATAC-seq (blue) on C/EBPα binding sites where chromatin accessibility is limited at 0 hr (n = 4550). Each panel depicts coverage at the indicated time. (**B**) Coverage plots of genome-wide data sets as indicated. Rows of all panels represent enhancers occupied by C/EBPα, sorted by eRNA synthesis kinetics and intensity. The coverage was aligned to the transcription start site (TSS) of eRNAs. (**C**) Log2-transformed counts of chromatin accessibility and eRNA synthesis on enhancers within chromatin regions previously inaccessible at 0 hr. (**D**) Difference between changes of chromatin accessibility and changes in eRNA synthesis at each locus (y-axis) between indicated time points (x-axis). Log2-transformed counts of chromatin accessibility and eRNA synthesis on enhancers within chromatin regions previously inaccessible at 0 hr. (**E**) Coverage tracks of eRNA synthesis (green) and chromatin accessibility (blue) over indicated time points illustrate that chromatin becomes accessible before enhancer transcription at *KDM7A* and *CD14* enhancers.

The online version of this article includes the following figure supplement(s) for figure 3:

**Figure supplement 1.** C/EBPα binding induces chromatin opening and activate enhancers.

events at putative enhancers during transdifferentiation. The results are consistent with a pioneering role of C/EBPα, which is able to invade chromatin at de novo enhancers and to induce chromatin opening, which later leads to eRNA synthesis and C/EBPα release at many of these sites.

## Multiple enhancers can be paired with target genes

We next aimed at identifying the target promoters for the enhancers. We paired all 7624 enhancers with potential target promoters using two different approaches (Materials and methods, *Figure 4A*). The 'neighboring approach' pairs enhancers with the nearest upstream or downstream promoters that were active at one time point at least. Enhancer–promoter pairs with significant changes in mRNA synthesis were further analyzed (*Figure 4B* and *Figure 4—figure supplement 1A*). Changes in eRNA synthesis correlated positively with changes in mRNA synthesis (*Figure 4—figure supplement 1C and D*), in agreement with what we observed in our previous analysis of T-cell activation (*Michel et al., 2017*). The '1 Mb method' pairs all enhancers that lie within 1 Mb, the median width of a human topologically associating domain (TAD) (*Dixon et al., 2012*; *Yan et al., 2017*), of a promoter if the correlation coefficient between synthesis of eRNA and mRNA was greater than 0.4

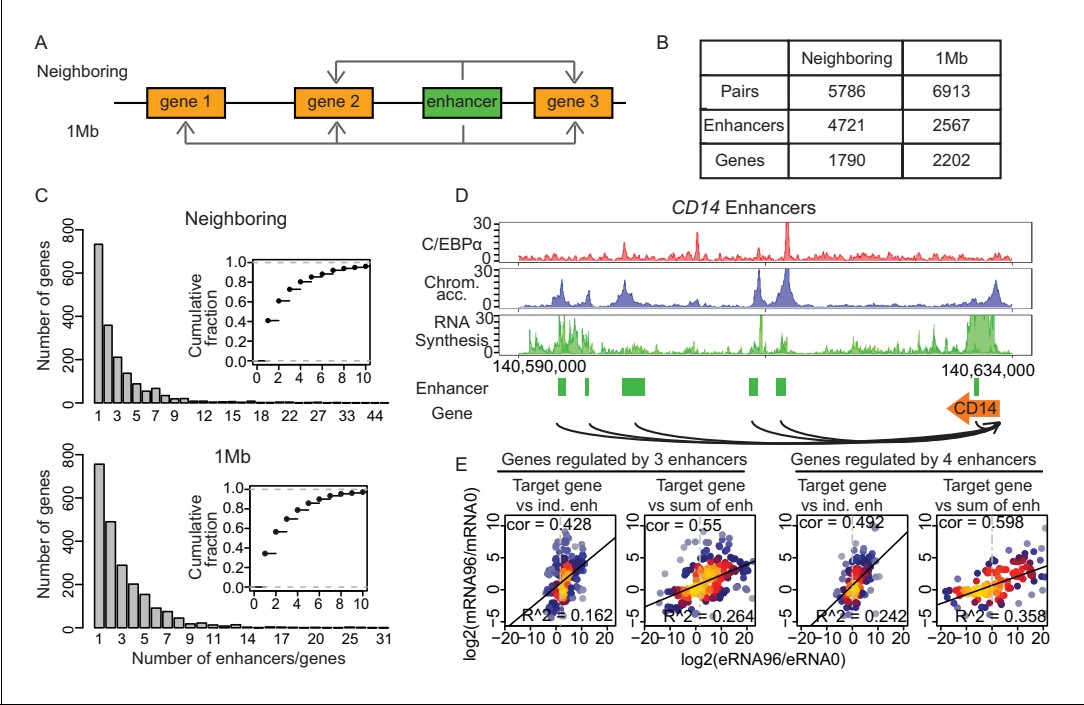

**Figure 4.** Multiple enhancers activate target gene promoters. (**A**) Schematic diagram describing 'neighboring' and '1 Mb' enhancer–promoter pairing methods. With the neighboring method, enhancers are paired to the nearest upstream and downstream transcribed promoters. Pairs with differentially regulated promoters were taken for further analysis. With the 1 Mb method, enhancers were paired with all promoters within 1 Mb, which corresponds to the median width of a human topologically associating domain (TAD) (*Dixon et al., 2012*; *Yan et al., 2017*). Pairs with promoters and enhancers that are differentially regulated in a correlated manner were analyzed further (Materials and methods). (**B**) Number of enhancer–promoter pairs, enhancers and genes per pair and method. (**C**) 59.1% and 65.7% of genes are regulated by more than one enhancer when using the neighboring and 1 Mb method, respectively. (**D**) Multiple enhancer cooperation at *CD14* gene is exemplified by the coverage tracks of indicated data sets at 96 hr. *CD14* gene is regulated by six enhancers. BRD4, H3K4me1, and H3K27Ac coverage tracks at 0 hr and 96 hr are depicted in *Figure 4—figure supplement 1E*. (**E**) Scatterplot of log2 fold change of mRNA synthesis level at 96 hr compared to 0 hr and log2 fold change of eRNA synthesis from individual (ind.) or sum of all enhancers. Genes regulated by three enhancers (left) or by four enhancers (right) are shown. Spearman's correlation coefficients at the top left corner and adjusted R-squared ($R^2$) values at the bottom right corner indicate that the sum of enhancer activity changes explains changes in mRNA synthesis better.

The online version of this article includes the following figure supplement(s) for figure 4:

**Figure supplement 1.** Enhancer–promoter pairing.

**Figure supplement 2.** Multiple enhancers activate target gene promoters.

(*Figure 4A*). Only pairs with significant changes in both eRNA and mRNA synthesis were further analyzed (*Figure 4B* and *Figure 4—figure supplement 1B*).

We found that 59.1% or 65.7% of putative target genes (1057 and 1446 genes, respectively) were paired with more than one enhancer when we used the neighboring or 1 Mb pairs, respectively, as exemplified for *CD14* (*Figure 4C and D*, *Figure 4—figure supplement 1E*). Obtaining similar results in parallel analyses with two different pairing methods strengthens our conclusions. To gain additional support for the promoter–enhancer pairing, we tested whether the pairs were located within the same TAD obtained from Hi-C data (*Stik et al., 2020*; *Phanstiel et al., 2017*). This showed that 98.3% of the promoter–enhancer pairs obtained by the neighboring method were found within the same TAD, and 96.2% of the pairs obtained with 1 Mb method were found within the same TAD at one or more time points (*Figure 4—figure supplement 1F*), as exemplified by *CD14* and *LMO2*, (*Figure 4—figure supplement 1G and H*). In summary, we identified putative target genes for the majority of enhancers and found that many genes can be paired with multiple enhancers, in line with previous reports (*Beagrie et al., 2017*; *Rubin et al., 2017*).

## Multiple enhancers cooperate to activate target genes

To investigate the temporal changes in transcription activity of target genes that were paired with multiple enhancers, we used the amount of eRNA production as a proxy for enhancer activity. We first examined whether target gene transcription changes relative to 0 hr can be explained by the activity changes of a single enhancer or whether they can only be explained when multiple enhancers were taken into account. Generally, target genes paired with more than one enhancer showed higher mRNA synthesis (*Figure 4—figure supplement 2A*), consistent with published observations (*Rubin et al., 2017*). We sorted target genes by the number of their paired enhancers, and calculated regression coefficients of log2(FC) of eRNA synthesis (explanatory variable) against log2(FC) of mRNA synthesis (target variable) (*Figure 4—figure supplement 2B*). As the number of paired enhancers increased, the regression coefficient increased, showing that stronger changes in mRNA synthesis were detected when more paired, active enhancers were present (*Figure 4—figure supplement 2B*).

Next, we tested whether the changes in promoter activity can be explained by the cumulative changes of all paired enhancers. We calculated the correlation between the log2(FC)s in mRNA synthesis and the sum of eRNA synthesis of the paired enhancers. There was a clear trend that promoter activity changes are better explained by the sum of activity changes in paired enhancers (*Figure 4E*). Generally, the correlation between promoter and enhancer activity changes improved with the number of paired enhancers (*Figure 4—figure supplement 2C and D*), as did the coefficient of determination ($R^2$) of a linear regression, and this was independent of the method of promoter–enhancer pairing (*Figure 4—figure supplement 2E and F*). These results indicate that multiple paired and active enhancers can contribute to target promoter activation and should be taken into account to understand changes in target gene transcription.

## Enhancer cooperation can be additive or synergistic

We next asked whether the contributions of individual enhancers to target gene activation are always additive or whether they can also be non-additive. This requires an analysis at the level of individual loci, which is only possible with time course data as available here. For robust curve fitting, we included TT-seq data for three additional time points, resulting in a total of seven time points (0, 12, 24, 30, 36, 72, and 96 hr). We fitted these data to three models (*Dukler et al., 2017*), an additive, a synergistic (or exponential), and a logistic model (*Figure 5A,B* and *Figure 5—figure supplement 1A*, respectively; Materials and methods). The additive model explains promoter activity with the sum of the activities of all enhancers paired with the neighboring method. The synergistic model assumes that changes in promoter activity over time are greater than what is predicted from the changes in the sum of the enhancer activities. The logistic model postulates that promoter activity reaches a limit and cannot increase any further even if the sum of the enhancer activities increases more, reflecting the known upper limit of promoter activity (*Gressel et al., 2017*; *Ikeda et al., 1992*). To determine the best-fitting model, we computed the Bayesian information criterion (BIC) score for each model and plotted the relative BIC, that is, the difference between the BIC score for the additive model and the BIC score of the tested models (*Figure 5—figure supplement 1B and C*).

To investigate which of the three models best explains transcription changes at target genes that are paired with multiple enhancers we estimated the relative BIC for 773 target genes that were paired with 2–20 enhancers according to the neighboring method. Of these target genes, 277, 92, and 250 were best described by the additive, synergistic, and logistic models, respectively (*Figure 5—figure supplement 1D*, Materials and methods). We ignored 154 genes with a relative BIC between 0 and 2, because they were ambiguous, showing reasonable fits to both additive and synergistic models (*Figure 5C*). For the 250 genes fitted with the logistic model, we observed that promoter activity depended on enhancer activity either in an additive or in a synergistic way before reaching a plateau (*Figure 5—figure supplement 1A*). To distinguish between these models, we excluded the data point with the highest sum of enhancer activities and calculated the BIC with the remaining six data points. Of the 250 genes, 71 and 44 genes could now be fitted with the additive or synergistic model, whereas 109 genes remained in the logistic class (*Figure 5—figure supplement 1E*).

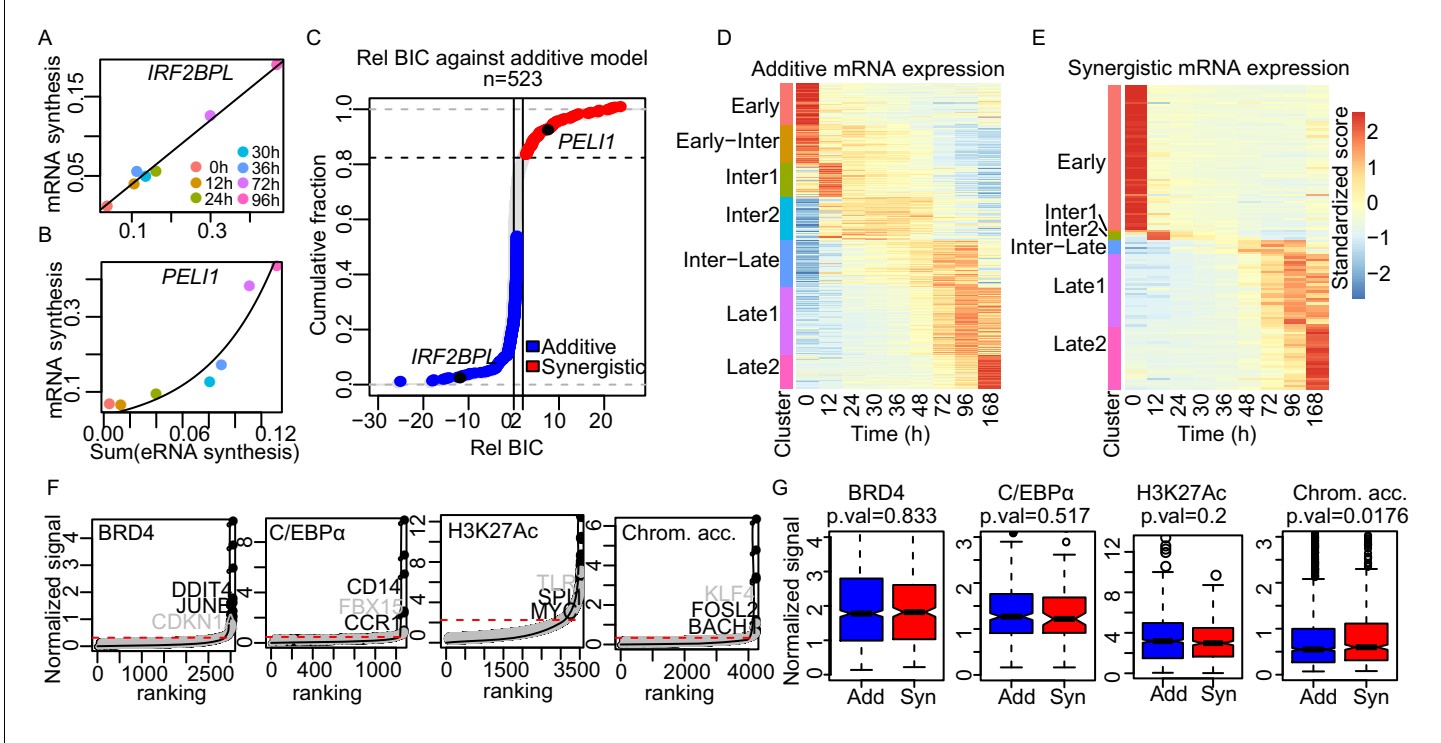

**Figure 5.** Enhancer cooperation can be additive or synergistic. (**A and B**) Exemplary curve fits of the additive (**A**) and the synergistic models (**B**) to the observed size factor and length normalized TT-seq RNA synthesis level for the indicated mRNA and the sum of the paired eRNAs. (**C**) The relative Bayesian Information Criterion (BIC) for the curve fits for a total of 523 promoters reveals that RNA synthesis at most loci follows either the additive or the synergistic model. The synergistic model fitted better for 17% of tested loci, showing a relative BIC greater than 2 (red). Any loci with a relative BIC between 0 and 2 were left as ambiguous because they fitted both models (gray). (**D and E**) Heatmap of gene expression (RNA-seq) regulated in additive (n = 348) (**D**) or synergistic (n = 136) (**E**) manner. Clusters are as shown in *Figure 1C* (k-means). (**F**) ChIP signal for BRD4, C/EBPα, and H3K27Ac, and ATAC-seq signal, ranked by signal strength on the peaks within 1 kb of our enhancers (Materials and methods, black dots) with cut-off values for superenhancers (red dashed lines). Synergistic enhancers are depicted with gray dots. (**G**) Additive (blue) and synergistic (red) enhancers are compared for their occupancy with BRD4, C/EBPα, H3K27Ac, or their chromatin accessibility (ATAC-seq). The comparison was carried out for the time point where the highest signal was observed. Fold differences in median signal of BRD4, C/EBPα, H3K27Ac, and chromatin accessibility between additive and synergistic enhancers are 1.02, 0.96, 1.07, and 1.09, respectively.

The online version of this article includes the following figure supplement(s) for figure 5:

**Figure supplement 1.** Enhancer cooperation can be additive or synergistic.

**Figure supplement 2.** Enhancer synergy is a robust phenomenon.

**Figure supplement 3.** Synergistic enhancers drive cell type-specific genes.

**Figure supplement 4.** Synergistic enhancers drive cell type-specific genes.

Taken together, our analysis indicates how enhancers cooperate to drive gene transcription over time. A total of 348 genes (~45%) were regulated in an additive manner by multiple enhancers. These genes included *Myc*, which is known to be regulated in an additive manner (*Bahr et al., 2018*). We also found a total of 136 genes (92 + 44, ~17%) that were regulated by multiple enhancers in a synergistic manner. These genes showed an exponential change in mRNA synthesis over time relative to the sum of eRNA synthesis from paired enhancers (*Figure 5B*), and included macrophage-related genes such as *CITED2, LYZ* (*Keshav et al., 1991*; *Kranc et al., 2009*), and B cell-related genes such as *BCL7A, LEF1*, and *TLE1* (*Haddad et al., 2004*). In summary, enhancers cooperate in an additive or a synergistic manner to regulate their target genes during the transdifferentiation process.

## Enhancer synergy is a robust phenomenon

To investigate the phenomenon of synergistic enhancer activity further, we carried out additional analyses. To exclude that the observed synergistic behavior is a consequence of a noisy relative BIC

distribution, we computed the relative BIC of an additive model with respect to a logarithmic model, which is not meaningful (*Figure 5—figure supplement 2A*). Strikingly, data from only three genes could be fitted with the logarithmic model, providing a negative control for our curve fitting. Next, we checked whether we missed distant enhancers that contribute to target gene activation. We carried out the analysis with enhancer–promoter pairs obtained with the 1 Mb pairing method. We observed 603 and 194 genes regulated by additive and synergistic enhancers, respectively (*Figure 5—figure supplement 2B*). The synergistically regulated genes found with the two different pairing methods strongly overlapped. Of the 57 synergistically regulated genes that were found with the neighboring method and could be tested, 33 were classified as synergistically regulated with the 1 Mb pairing method (Fisher's exact test p-value = 3.5e-11) (*Figure 5—figure supplement 2C*).

Since the number of enhancers paired to each gene varies between the neighboring and 1 Mb methods, only a small number of genes could be tested for enhancer synergy. We therefore used the third method of promoter–enhancer pairing referred to as 'TAD pairing'. TAD pairing pairs all promoters and enhancers that lie within a TAD that was experimentally identified with Hi-C data (*Stik et al., 2020*). We assumed pairing if the correlation coefficient between changes in synthesis of eRNA and mRNA was greater than 0.4. Pairs with significant changes in both eRNA and mRNA synthesis were further analyzed. We found 7355 pairs with 1882 promoters and 2567 enhancers. Similar to other pairing methods (*Figure 4C*), 69% of putative target genes (1296 genes) were paired with more than one enhancer (*Figure 5—figure supplement 2D*). We observed 561 and 238 genes regulated by additive or synergistic enhancers, respectively. Out of 68 synergistically regulated genes that were found with the neighboring method and could be tested, 52 were also classified as synergistically regulated with TAD pairing (Fisher's exact test p-value = 2.9e-19) (*Figure 5—figure supplement 2E*). Thus, the synergistically regulated genes found with the different pairing approaches overlapped significantly, providing further support of our findings.

We further tested whether we had underestimated the number of enhancers because this could lead to incorrect assignment of synergistic behavior. We extended our list of putative enhancers by including 7863 additional merged ncRNA TUs that lacked H3K4me1 or ATAC-seq signals. Using the neighboring method, this increased the number of paired enhancers to 1184 out of 1790 TUs. We found that 97 of the 136 previously identified synergistically regulated genes (71%) were again detected to be regulated in a synergistic manner (*Figure 5—figure supplement 2F*). This indicates that the widespread synergistic behavior we observed was not due to an underestimation of the number of enhancers. In order to check whether the observed enhancer synergy was due to the underestimation of enhancer activities, we simulated an increase of eRNA transcription and determined synergistically regulated genes. Synergistically regulated genes were not affected by the artificial increase of eRNA transcription (*Figure 5—figure supplement 2G*). We further found no difference in the number of enhancers between additive and synergistic loci (median 3.5 and 3, respectively) (*Figure 5—figure supplement 2H*).

We also asked whether one highly active enhancer could dominate gene activation for synergistic enhancers. This was not the case because the median contribution from the strongest enhancer to the total eRNA signal was the same for additive and synergistic enhancers (*Figure 5—figure supplement 2I*). As an additional control, we asked whether synergistic enhancers could regulate two target genes, and whether in this case both target genes are regulated in the same manner. We found that 694 additive or synergistic enhancers were paired with two target gene promoters using the neighboring method. Of these, 463 enhancers were paired with target genes that were either both regulated in an additive manner (463 enhancers, 254 promoters) or both in a synergistic manner (139 enhancers, 84 promoters) (*Figure 5—figure supplement 2J*, binomial p.val <2.2e-16). This is consistent with coordinated transcriptional bursting of two promoters regulated by one enhancer (*Fukaya et al., 2016*). These controls and analyses confirm the observed enhancer synergy and indicate that co-operative action is a property of enhancers, not promoters.

## Synergistic enhancers are involved in cell type-specific gene expression

We next investigated the nature of the target genes of synergistic enhancers. We found that most target genes of synergistic enhancers (85%) were cell type-specific genes, that is, genes that were specifically expressed either in B cells (60 TUs) or in macrophage-like cells (56 TUs) according to RNA-seq data (*Figure 5D and E*). B cell-specific genes controlled by synergistic enhancers include *MYB, BLNK, VPREB1,* and *IGLL5* (*Cobaleda et al., 2007*; *Thomas et al., 2005*). Macrophage-specific

genes regulated by synergistic enhancers include *CEBPB, VSIG4,* and *ITGAX* (*Friedman, 2007*; *Lavin et al., 2014*; *Vogt et al., 2006*). Macrophage-specific, synergistically regulated genes often remained inactive until 36 hr, while additive genes showed activation from 12 hr onwards (*Figure 5— figure supplement 3A*). Gene ontology analysis of the genes regulated by additive and synergistic enhancers confirmed the largely cell type-specific nature of the latter ones (*Figure 5—figure supplement 3B*). Thus, the majority of synergistic enhancers drive cell type-specific genes.

Further analysis showed that the delayed but rapid induction of synergistically regulated genes was not a consequence of late chromatin opening. Although the average chromatin accessibility changed more at synergistic enhancers compared to additive enhancers (45% vs 38% of overlapping ATAC-seq peaks, Fisher's exact test p-value = 0.01), the kinetics of chromatin opening were similar for both types of enhancers at late upregulated genes (*Figure 5—figure supplement 3C*). The initial chromatin accessibility at 0 hr was also comparable for genes regulated by additive or synergistic enhancers (*Figure 5—figure supplement 3D*). Thus, there is no evidence that enhancer synergy at target genes that are strongly induced at later time points is simply a consequence of delayed chromatin opening at these enhancers. Instead, these observations point to another mechanistic basis for enhancer synergy.

We further tested whether our synergistic enhancers may be described as 'superenhancers', because these can also target cell type-specific genes. Superenhancers are generally defined by their high occupancy with specific TFs and coactivators (*Hnisz et al., 2013*). We collected ChIP-seq data for the Mediator subunit MED1, the chromatin regulator BRD4 (Materials and methods), and used these and recent ChIP-seq data for H3K27Ac (*Stik et al., 2020*) for superenhancer calling. The MED1 signal was weak and did not allow for a robust analysis. However, the BRD4 and H3K27Ac signal could be used to generate a typical 'superenhancer plot' (*Hnisz et al., 2013*; *Sabari et al., 2018*; *Whyte et al., 2013*; *Figure 5F*). We also generated such plots for C/EBPα ChIP-seq data and for the ATAC-seq data. All four plots identified enhancers with high signals (*Figure 5F*, *Figure 5—figure supplement 4A and B*). Although the superenhancers derived this way show high occupancy or accessibility at 96 hr, RNA synthesis from these superenhancers and their paired promoters was not always the highest at 96 hr compared to other time points (*Figure 5—figure supplement 4C*). Consistent with this, the genes paired with superenhancers included not only cell type-specific genes such as CD14, DDIT4, JUNB, and FOSL2, but also genes that are commonly expressed across different cell types, such as DNMT1, ARID5B, and MAP3K1. Gene ontology analysis on the paired genes of superenhancers includes cell type-specific GO terms such as immune response and inflammatory response as well as general GO terms such as regulation of cellular process and cell–cell adhesion (*Supplementary files 2–5*).

Furthermore, the superenhancers detected by occupancy or accessibility signals did not correspond to the synergistic enhancers we identified here based on RNA synthesis changes. Indeed, previously identified superenhancers from the dbSUPER database (*Hnisz et al., 2013*; *Khan and Zhang, 2016*) overlapped to similar extents with our additive or synergistic enhancers (*Figure 5—figure supplement 4D*). Additionally, there was no difference in the occupancy signals between synergistic enhancers and additive enhancers (*Figure 5G*). In conclusion, the synergistic enhancers we identified based on functional data (RNA synthesis) are involved in cell type-specific gene expression and do not generally correspond to classically defined superenhancers.

## Synergistic enhancers are regulated by cell type-specific TFs

We finally investigated whether enhancer synergy depends on binding of specific TFs to enhancer regions. According to our RNA-seq data, different sets of TFs are expressed in cascades from 0 hr to 96 hr during transdifferentiation (*Figure 6—figure supplement 1A*). We located sites in our enhancers that contained DNA motifs for these TFs. Then we determined which of these sites are occupied by TFs, using our ATAC-seq data to perform TF footprinting on the enhancers regulating late upregulated genes (*Sherwood et al., 2014*). This analysis revealed that motifs of 23 TF subfamilies are significantly enriched in our synergistic enhancers compared to the additive enhancers (Fisher's exact test p-value < 0.05, *Figure 6—figure supplement 1B*). These enriched TFs include C/EBP, PU.1, CREB1, ETS, KLF, and RUNT family factors (*Figure 6A*). TFs in these categories are known to function in macrophages (*Friedman, 2007*). In particular, C/EBPα/β/ε are required for granulocyte–monocyte progenitors (GMP) and macrophages (*Friedman, 2007*). Knockdown of PU.1 and C/EBPβ was shown to impair transdifferentiation (*van Oevelen et al., 2015*), and C/EBPβ dimerizes with

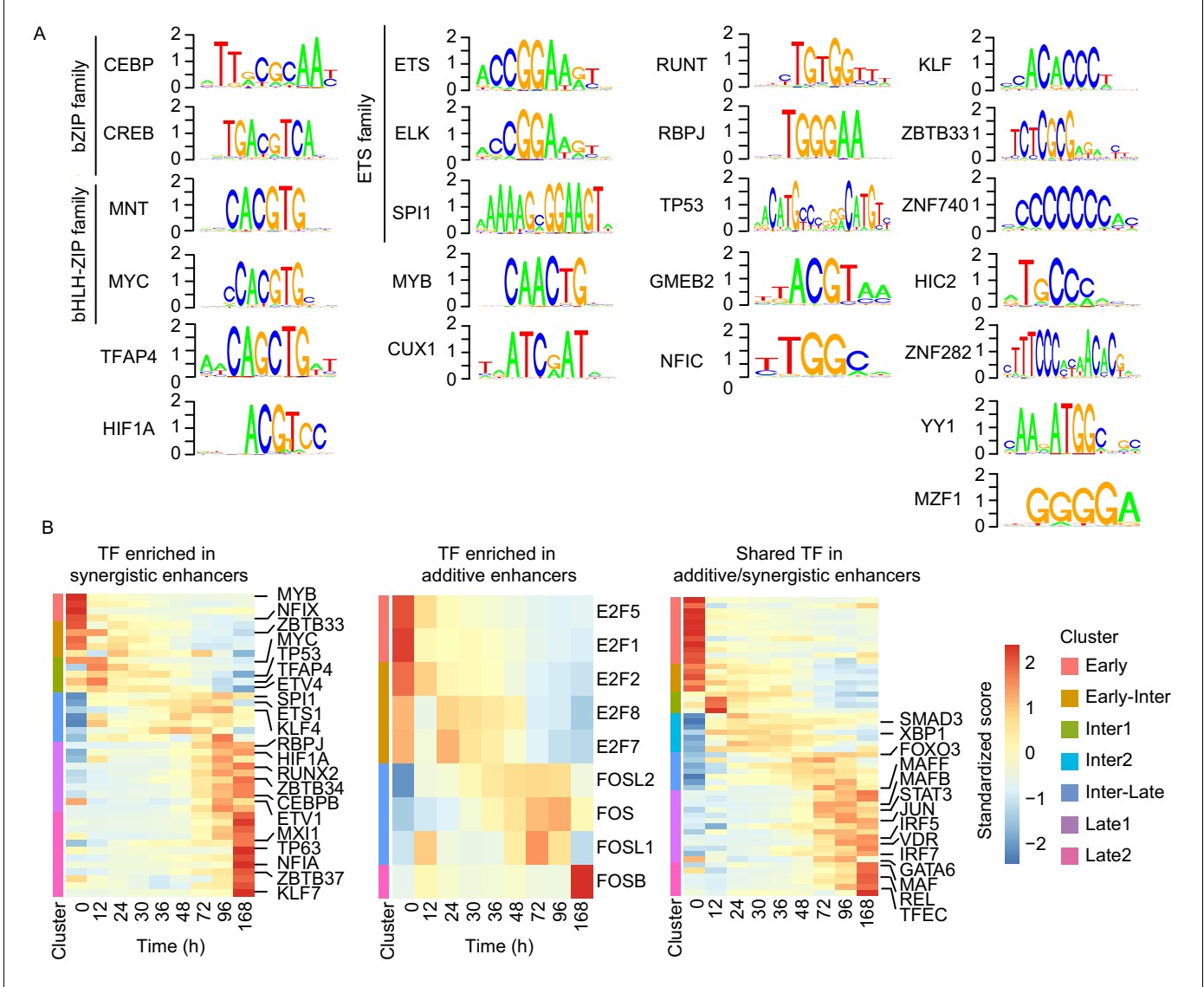

**Figure 6.** Cell type-specific transcription factors are enriched in synergistic enhancers. (**A**) Motifs of TFs enriched in synergistic enhancers from TF footprinting analysis (Materials and methods). (**B**) Expression of differentially expressed TFs (| log2(FC) |>1, FDR < 0.05, RNA-seq) enriched in synergistic (left panel), additive (middle panel), and both synergistic and additive (right panel) enhancers.

The online version of this article includes the following figure supplement(s) for figure 6:

**Figure supplement 1.** Specific sets of TFs are enriched on synergistic enhancers.

CREB1 to activate macrophage genes (*Ruffell et al., 2009*). Members of the KLF family, such as KLF1-4 and KLF6, are important for monocyte and macrophage activation (*Cao et al., 2010*; *Date et al., 2014*). C/EBPα, PU.1, and RUNX1 are frequently mutated in acute myeloid leukemia (AML), which can arise from reduced transcription activity and impede myeloid differentiation (*Rosenbauer and Tenen, 2007*).

In contrast, additive enhancers contained motifs of only two TF families that were enriched compared to synergistic enhancers, in particular the E2F and FOS (*Figure 6—figure supplement 1C*). TFs of the E2F family have general regulatory functions in macrophages, such as cell cycle and apoptosis regulation (*Trikha et al., 2011*). TFs of the FOS family also have regulatory functions in macrophages (*Friedman, 2007*). Binding to FOS motifs was enriched in additive enhancers only at 12 hr (*Figure 6—figure supplement 1B*), indicating that at later time points FOS family factors bind to additive and synergistic enhancers similarly.

In general, synergistic enhancers were enriched for macrophage-specific TFs, whereas additive enhancers were not. To confirm this, we investigated the expression of TFs that belong to the same subfamilies with the enriched TFs in additive or synergistic enhancers. Indeed, of the TFs that were enriched in synergistic enhancers and differentially expressed, about half were macrophage specific, whereas TFs of E2F family that were enriched in additive enhancers were not (*Figure 6B*). This suggests that additional TFs are needed to activate macrophage-specific additive genes during the late transition (96 hr) (*Figure 6B*). This is possibly due to binding of TFs at both additive and synergistic enhancers such as those for the IRF and MAF family factors that are also known to regulate macrophage genes (*Friedman, 2007*). Many of these shared TFs are indeed upregulated at 96 hr (*Figure 6B*). These results indicate that a set of shared TFs bind to both types of enhancers, whereas a specific set of macrophage-specific TFs additionally and preferentially binds to synergistic enhancers to drive macrophage-specific target gene promoters that establish the new cell type.

## Discussion

How enhancers cooperate to drive target gene expression and to determine cell types is a central question in the field of genomic regulation. However, the question of functional enhancer cooperativity could thus far not be addressed in a systematic manner in a dynamic system. A route to such an analysis has now been provided by the availability of an efficient and simple transdifferentiation system and the development of TT-seq to measure genome-wide RNA synthesis from both enhancers and target gene promoters. Here we investigated enhancer cooperation over time during transdifferentiation of human B-cells to macrophage-like cells. We examined this process in a quantitative manner, assuming eRNA synthesis to be a proxy for enhancer activity. We identified transcriptionally active enhancers, their temporal activity changes, and used correlations to predict how multiple enhancers cooperate to regulate their target genes. These analyses indicated that multiple enhancers often act in an additive manner, and that synergistic enhancer action occurs at ~20% of the tested putative target genes. Target genes of synergistic enhancers were almost exclusively cell type-specific.

The synergistic enhancers identified here were different from previously reported superenhancers. We identified synergistic enhancers from changes in RNA synthesis activity during the time course, whereas identification of superenhancers generally uses TF and coactivator occupancy measurements at one time point. Whereas our synergistic enhancers are not necessarily arranged next to each other, superenhancers are generally linear clusters of constituent enhancers. In contrast to our time-resolved study, previous studies investigated enhancer cooperation in steady state or at an end point, and at selected genes, rather than genome-wide (*Hay et al., 2016*; *Osterwalder et al., 2018*; *Stine et al., 2011*). It was suggested that superenhancers retain stable interactions via nuclear condensation that involves phase separation (*Hnisz et al., 2017*).

Although the molecular mechanisms underlying enhancer cooperation remain to be investigated, simple models may explain our observations (*Figure 7*). An additive effect of enhancers on gene expression is expected if the enhancers act independently of each other and do not need each other for productive transcription. This additive mode of regulation is predicted to allow for fine-tuning of promoter activity and to enable robust expression. It would better tolerate the loss of one of the enhancer activities, for instance due to mutations in TF-binding sites or due to changes in the expression levels of TFs that bind the enhancers. Indeed, expression of gap genes in *Drosophila* are regulated by multiple enhancers, and deleting one of these enhancers has little effect on the gene expression pattern (*Perry et al., 2011*). Thus, additive enhancers would be advantageous for genes that are expressed and function in multiple cell types, and indeed we observe that additive enhancers target genes commonly expressed throughout multiple transitions during transdifferentiation.

A synergistic effect of enhancers on target gene transcription is expected when the enhancers function together (*Figure 7*). Such enhancer cooperation could be simultaneous and may rely on cooperative nuclear condensation based on liquid–liquid phase separation (*Boehning et al., 2018*; *Boija et al., 2018*; *Chong et al., 2018*; *Hnisz et al., 2017*; *Murray et al., 2017*; *Nair et al., 2019*; *Sabari et al., 2018*). Indeed, some of the TFs with enriched motifs in our synergistic enhancers have been shown to undergo phase separation with Mediator (MYC and TP53) or to interact with Mediator (C/EBPβ) (*Boija et al., 2018*; *Li et al., 2008*). Nuclear condensation may also allow for a higher

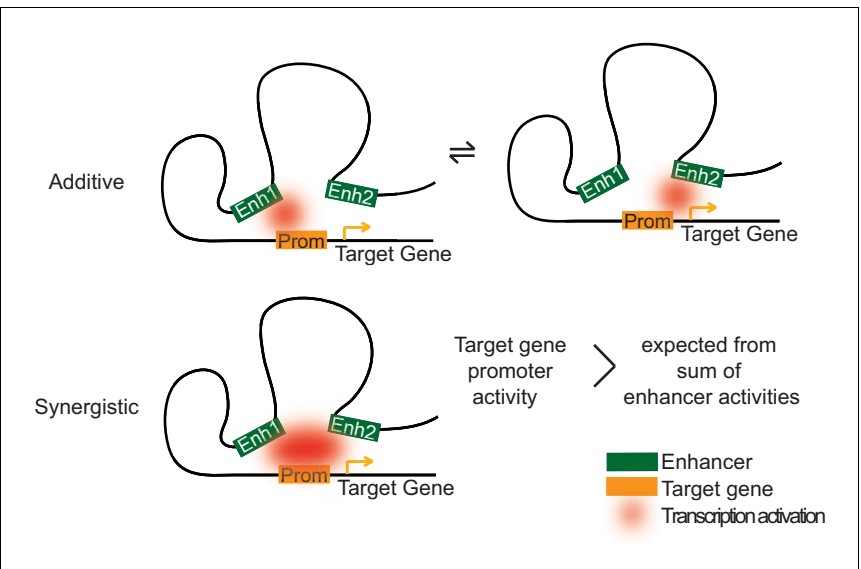

**Figure 7.** Two models of how multiple enhancers regulate target genes. Simple models may explain additive and synergistic enhancer cooperation during target gene activation. Additive cooperation may result from alternative activation of the target promoter by each constituent enhancer at different time points. Synergistic cooperation may be achieved if multiple enhancers contact the target gene promoter simultaneously. For details compare text.

frequency of transcriptional bursting (*Larsson et al., 2019*). Additionally or alternatively, enhancer synergy may also stem from sequential cooperation if different enhancers target different steps of transcription such as initiation and elongation (*Beagrie and Pombo, 2016*; *Haberle and Stark, 2018*; *Henriques et al., 2018*; *Kim et al., 1998*; *Sawado et al., 2003*; *Spicuglia et al., 2002*). The effect of synergistic enhancer cooperation on target gene activity is expected to be switch-like and would be beneficial for cell type-specific genes. Such switches would allow for efficient determination of cell types during differentiation, because genes that are active in the old cell type could be rapidly switched off and genes expressed in the new cell type could suddenly be switched on at a certain stage of differentiation.

Finally, our results improve our understanding of step-wise enhancer and gene activation during a transdifferentiation process. Previous work showed that a subset of TFs is critical for enhancer priming and chromatin remodeling (*Heinz et al., 2010*; *Takaku et al., 2016*). C/EBPα can prime enhancers and initiate chromatin opening by interactions with MLL3/4 complexes (*Lee et al., 2013*) and with SWI/SNF chromatin remodeling complexes (*Pedersen et al., 2001*). Consistent with this, we find that chromatin opening follows C/EBPα binding and that this results in subsequent enhancer transcription. These observations are consistent with the capacity of the pioneering factor C/EBPα to induce DNA opening (*Iberg-Badeaux et al., 2017*; *van Oevelen et al., 2015*), and its ability to reorganize chromatin states and genome architecture before gene expression changes (*Stadhouders et al., 2018*). It remains controversial whether enhancer transcription is required for enhancer priming (*Calo and Wysocka, 2013*; *Dorighi et al., 2017*; *Kaikkonen et al., 2013*), but our data support that in our system the chromatin modification H3K4me1 generally precedes nucleosome depletion and enhancer transcription (*Bonn et al., 2012*; *Creyghton et al., 2010*; *Rada-Iglesias et al., 2011*; *Wamstad et al., 2012*). Our results also explain how activation of the glucocorticoid receptor can induce chromatin decompaction in the presence of the transcription inhibitor α-amanitin (*Jubb et al., 2017*).

In conclusion, our data reveal the stepwise activation of enhancers and genes during transdifferentiation following binding of the pioneering factor C/EBPα. We provide evidence from time course analysis that enhancers can cooperate in an additive or synergistic manner to alter the activity of target genes. Synergistic enhancers tend to bind cell type-specific TFs and regulate cell type-specific genes, and we speculate that this enables a switch-like expression behavior for such genes. The synergistic enhancers identified here are generally distinct from previously reported superenhancers.

Finally, our approach can be used to investigate enhancer cooperation in many other cellular processes.

# Materials and methods

## Key resources table

| Reagent type (species) or resource | Designation | Source or reference | Identifiers | Additional information |
|---|---|---|---|---|
| Cell line (*Homo sapiens*; female) | BLaER1 B-cell precursor leukemia | Laboratory of Thomas Graf | RRID:CVCL_VQ57 | RCH-ACV stably expressing estrogen inducible C/EBPα |
| Commercial assay, kit | Plasmo Test Mycoplasma Detection Kit | InvivoGen | rep-pt1 | |
| Commercial assay, kit | NUGEN Ovation V2 Kit | NUGEN | 0343 | |
| Commercial assay, kit | NEB Ultra DNA Library kit | NEB | E7370S | |
| Commercial assay, kit | KAPA Real-Time Library Amplification Kit | Peqlab | KK2701 | |
| Commercial assay, kit | Nextera Tn5 Transposase | Illumina | FC-121–1030 | |
| Chemical compound, drug | Human CSF-1 | PEPROTECH | 300–25 | |
| Chemical compound, drug | Human IL-3 | PEPROTECH | 200–03 | |
| Chemical compound, drug | β-estradiol | CALBIOCHEM | 3301 | |
| Chemical compound, drug | 4-thiouridine | Carbosynth | 13957-31-8 | |
| Antibody | Anti-C/EBPα (rabbit polyclonal) | Santa Cruz | Cat# sc-61, RRID:AB_631233 | ChIP-seq (5 µg for 50 µg of chromatin) |
| Antibody | Anti-H3K4me1 (rabbit polyclonal) | Abcam | Cat# ab8895, RRID:AB_306847 | ChIP-seq (5 µg for 30 µg of chromatin) |
| Antibody | Anti-BRD4 (rabbit polyclonal) | Bethyl Laboratories | Cat# A301-985A100, RRID:AB_2620184 | ChIP-seq (5 µg for 100 µg of chromatin) |
| Antibody | Anti-human CD19-APC-cy7APC-cy7 mouse anti-human CD19 (mouse monoclonal) | BD Biosciences | Cat# 557791, RRID:AB_396873 | FACS (2.5 µL per test) |
| Antibody | Anti-human CD14-PEPE mouse anti-human CD14 (mouse monoclonal) | BD Biosciences | Cat# 555398, RRID:AB_395799 | FACS (5 µL per test) |
| Sequenced-based reagent | CD19 forward | *Rapino et al., 2013* | qPCR primers | GATGCAGACTCTTATGAGAAC |
| Sequenced-based reagent | CD19 reverse | *Rapino et al., 2013* | qPCR primers | TCAGATTTCAGAGTCAGGTG |
| Sequenced-based reagent | IGJ forward | *Rapino et al., 2013* | qPCR primers | TGTTCATGTGAAAGCCCAAG |
| Sequenced-based reagent | IGJ reverse | *Rapino et al., 2013* | qPCR primers | TCGGATGTTTCTCTCCACAA |
| Sequenced-based reagent | VPREB3 forward | *Rapino et al., 2013* | qPCR primers | GGGGACCTTCCTGTCAGTTT |
| Sequenced-based reagent | VPREB3 reverse | *Rapino et al., 2013* | qPCR primers | ACCGTAGTCCCTGATGGTGA |

*Continued on next page*

*Continued*

| Reagent type (species) or resource | Designation | Source or reference | Identifiers | Additional information |
|---|---|---|---|---|
| Sequenced-based reagent | CD14 forward | *Rapino et al., 2013* | qPCR primers | GATTACATAAACTGTCAGAGGC |
| Sequenced-based reagent | CD14 reverse | *Rapino et al., 2013* | qPCR primers | TCCATGGTCGATAAGTCTTC |
| Sequenced-based reagent | FCGR1B forward | *Rapino et al., 2013* | qPCR primers | CCTTGAGGTGTCATGCGTG |
| Sequenced-based reagent | FCGR1B reverse | *Rapino et al., 2013* | qPCR primers | AAGGCTTTGCCATTTCGATAGT |
| Sequenced-based reagent | ITGAM forward | *Rapino et al., 2013* | qPCR primers | GGGGTCTCCACTAAATATCTC |
| Sequenced-based reagent | ITGAM reverse | *Rapino et al., 2013* | qPCR primers | CTGACCTGATATTGATGCTG |
| Sequenced-based reagent | GAPDH forward | This paper | qPCR primers | TCTCTGCTCCTCCTGTTCGAC |
| Sequenced-based reagent | GAPDH reverse | This paper | qPCR primers | GGCGCCCAATACGACCAAAT |
| Software, algorithm | Cutadapt | *Martin, 2012* | RRID:SCR_011841 | Version 1.9.1 |
| Software, algorithm | Trim Galore | https://www.bioinformatics.babraham.ac.uk/projects/trim_galore/ | RRID:SCR_011847 | Version 0.4.1 |
| Software, algorithm | FastQC | http://www.bioinformatics.bbsrc.ac.uk/projects/fastqc | RRID:SCR_014583 | |
| Software, algorithm | SAMTOOLS | *Li et al., 2009* | RRID:SCR_002105 | Version 1.2 |
| Software, algorithm | STAR | *Dobin et al., 2013* | RRID:SCR_004463 | Version 2.4.2 |
| Software, algorithm | DESeq2 | *Love et al., 2014* | RRID:SCR_015687 | |
| Software, algorithm | Bowtie 2 | *Langmead and Salzberg, 2012* | RRID:SCR_016368 | Version 2.2.5 |
| Software, algorithm | MACS | *Zhang et al., 2008* | RRID:SCR_013291 | Version 2.1.1 |
| Software, algorithm | Bowtie | *Langmead et al., 2009* | RRID:SCR_005476 | Version 1.0.0 |
| Software, algorithm | HTSeq | *Anders et al., 2015* | RRID:SCR_005514 | Version 0.6.1p1 |
| Software, algorithm | ggplot2 | http://ggplot2.org | RRID:SCR_014601 | |
| Software, algorithm | Pheatmap | https://CRAN.R-project.org/package=pheatmap | RRID:SCR_016418 | |
| Software, algorithm | DAVID | *Huang et al., 2009* | RRID:SCR_001881 | |
| Software, algorithm | GenoSTAN | *Zacher et al., 2017* | | |
| Software, algorithm | MEME Suite - motif-based sequence analysis tools | *Bailey et al., 2009* | RRID:SCR_001783 | Version 4.11.2 |
| Software, algorithm | 3D Genome | http://promoter.bx.psu.edu/hi-c/ | RRID:SCR_017525 | |

*Continued on next page*

*Continued*

| Reagent type (species) or resource | Designation | Source or reference | Identifiers | Additional information |
|---|---|---|---|---|
| Software, algorithm | DEoptim | *Dukler et al., 2017* | | |
| Software, algorithm | ROSE | *Lovén et al., 2013*; *Whyte et al., 2013* | RRID:SCR_017390 | |
| Software, algorithm | PIQ | *Sherwood et al., 2014* | | |
| Software, algorithm | TFBSTool | *Tan and Lenhard, 2016* | | |

## Cell line

We used BLaER1 cells derived from RCH-ACV, precursor leukemia B cells, stably expressing estrogen-inducible C/EBPα (*Rapino et al., 2013*); these cells contain the Squirrel Monkey Retrovirus (SMRV). Cells were cultured in RPMI 1640 (GIBCO, 31870–025) with 10% FBS (GIBCO, 10500–064), 2% Glutamax (GIBCO, 35050–038), 2% Penicillin/Streptomycin (GIBCO, 15140122), and HEPES (GIBCO, 15630–056) at 37°C under 5% $CO_2$. Cells were regularly checked and tested negative for mycoplasma infection using Plasmo Test Mycoplasma Detection Kit (InvivoGen, rep-pt1). For our time course experiments, cells were treated with beta estradiol (100 nM) (CALBIOCHEM, 3301), human IL-3 (10 ng/ml) and human CSF-1 (10 ng/ml) (PEPROTECH, 200–03 and 300–25). Cells were harvested at various time points after treatment (0, 12, 24, 30, 36, 48, 72, 96, and 168 hr).

## RNA-seq and TT-seq

RNA-seq and TT-seq were performed from two biological replicates at nine and seven time points, respectively. At each time point, $1–2.2 \times 10^8$ cells were labeled with 0.5 mM 4-thiouridine (4sU, Carbosynth, 13957-31-8) for 5 min and harvested by centrifuging at 500 g for 2 min. Harvested pellets were lysed with QIAzol (Qiagen, 79306). We generated RNA spike-ins based on sequences of six ERCC RNA spike-in mix – ERCC-00043, ERCC-00170, ERCC-00136, ERCC-00145, ERCC-00092, and ERCC-00002, and prepared spike-in mix as described previously (*Huang et al., 2009*). Following addition of 5 ng of spike-in mix per $1 \times 10^8$ cells, total RNA was purified using QIAzol (Qiagen, 79306) according to the manufacturer's instructions. Total RNA (300 μg) was fragmented to 1500–5000 bp in size using Covaris S220 Ultrasonicator. Fraction of total RNA was set aside for total-RNA-seq. Newly synthesized RNAs were purified as described (*Schwalb et al., 2016*) with the following modifications. After streptavidin pull down, the resulting RNA was purified with RNeasy Micro Kit (Qiagen, 74004), together with DNase treatment (Qiagen, 79254). Libraries were generated with Nugen Ovation Universal RNA-seq System (Nugen, 0343), and amplified by nine cycles in addition to two initial cycles according to the manufacturer's instructions. Libraries were sequenced at $2 \times 50$ bp on HiSeq2000 (TAL, Göttingen) or at $2 \times 75$ bp on an Illumina NextSeq machine in house. The average read numbers obtained for total RNA-seq were $1 \times 10^7$ and $2 \times 10^8$ for TT-seq.

## Chromatin immunoprecipitation sequencing (ChIP-seq)

ChIP-seq was performed from two biological replicates at four time points. At each time point, $3 \times 10^7$ cells were harvested and fixed with 1% of formaldehyde for 8 min (Thermo Scientific, 28908). Formaldehyde was quenched with 0.2 M glycine for 5 min. Chromatin was sheared to 200–500 bp in size using the Covaris S220 ultrasonicator. A fraction of the resulting chromatin was used as input control. Sheared chromatin (50 μg for C/EBPα, 30 μg for H3K4me1, 100 μg for BRD4) was incubated with 5 μg of antibodies – anti-C/EBPα (Santa Cruz, sc-61), anti-H3K4me1 (abcam, ab8895), and anti-BRD4 (Bethyl Laboratories, A301–985A100) – bound to protein-A beads (Life Technologies, 9999–01). Samples were incubated at 65°C overnight to reverse the crosslinks, followed by RNase and proteinase K treatments. Purified ChIP and input DNA were quantified by Qubit 2.0 Fluorometer (Life Technologies, Q32866). Libraries were generated using NEB Ultra DNA Library kit (NEB, E7370S) and amplified by nine cycles with KAPA Real-Time Library Amplification Kit (Peqlab, KK2701). The libraries of BRD4-ChIP-seq were sequenced at $2 \times 75$ bp on an Illumina NextSeq machine in house.

The other libraries were sequenced at 2 × 50 bp on HiSeq2000 (TAL, Göttingen or LAFUGA, LMU München). The average read numbers obtained were $4 \times 10^7$ for C/EBPα ChIP-seq, $7 \times 10^7$ for H3K4me1 ChIP-seq, and $2 \times 10^7$ for BRD4 ChIP-seq.

## ATAC-seq

ATAC-seq was performed from two biological replicates at four time points as described (*Buenrostro et al., 2013*). Briefly, $5 \times 10^4$ cells at each time point were harvested and treated with Nextera Tn5 Transposase (Illumina, FC-121–1030) for 45 min at 37°C. Library fragments were amplified using 1× NEBNext High-Fidelity 2× PCR Master Mix (NEB, M0541S) and 1.25 µM of custom Nextera PCR primers 1 and 2. PCR amplification was done with 11 cycles, determined by KAPA Real-Time Library Amplication Kit (Peqlab, KK2701) to stop prior to saturation. Then, the samples were purified using Qiagen MinElute PCR Purification Kit (Qiagen, 28004) and with Agencourt AMPure XP beads (Beckman Coulter, A63881) in 3:1 ratio. The libraries were sequenced at 2 × 50 bp on HiSeq2000 (TAL, Göttingen). The average read numbers obtained were $3 \times 10^7$.

## Fluorescence-activated cell sorting (FACS)

Cells were harvested, washed with PBS, and blocked with Human FC Receptor Binding Inhibitor (ThermoFisher, 16–9161). Cells were incubated with antibodies, APC-cy7 Mouse Anti-Human CD19 (BD Pharmingen, 557791), PE Mouse Anti-Human CD14 (BD Pharmingen, 555398). After washing with PBS, cells were resuspended in 7AAD (ThermoFisher, 00–6993) diluted with PBS. The analysis was performed on BD Accuri C6 flow cytometer (BD Biosciences) according to standard protocols with following changes in filters. We used filter 675/25 at F3 and 670/LP at F4.

## Quantitative RT-PCR (qRT-PCR)

Cells were harvested with QIAzol and purified using RNeasy Mini Kit (Qiagen, 74104). The purified RNA was quantified using a Nanodrop spectrophotometer (Thermofisher), and cDNA was generated by reverse transcription PCR. The resulting cDNA was used as a template for quantitative PCR (qPCR, SybrGreen). Ct values were normalized with *GAPDH* expression. Relative expression levels were calculated at each time points compared to 0 hr for B cell markers or to 168 hr for macrophage markers. Primers used for qRT-PCR are listed in *Supplementary file 6*. Error bars represent standard deviation from three biological replicates.

## Quantification and statistical analysis

All sequencing data were trimmed using Cutadapt 1.9.1 (*Martin, 2012*) and trim_galore_v0.4.1 (https://www.bioinformatics.babraham.ac.uk/projects/trim_galore/). Read quality was screened using FastQC (http://www.bioinformatics.bbsrc.ac.uk/projects/fastqc). After alignments, Samtools1.2 (*Li et al., 2009*) was used to filter the alignments with MAPQ smaller than 7 (-q 7), and to retain only the proper pairs (-f99, -f147, -f83, -f163). Boxplots show inter-quantile ranges, with the median indicated with black line. Whiskers of boxplots represent maximum and minimum values excluding outliers; outliers were calculated as values greater or lower than 1.5 times the interquartile range. Significance was tested by Wilcoxon test, unless otherwise stated.

## RNA-seq and TT-seq data processing

Trimmed reads were aligned to GRCh38 genome assembly (Human Genome Reference Consortium) using STAR2.4.2 (*Dobin et al., 2013*). For coverage profiles and visualization, reads were uniquely mapped, antisense corrected, and normalized with size factors calculated from DESeq2 (*Love et al., 2014*; *Michel et al., 2017*). Spearman's correlation coefficients between replicates were 0.99 for total RNA-seq and TT-seq on protein coding genes at all time points.

## ChIP-seq data processing

Trimmed reads were aligned to GRCh38 genome assembly using Bowtie2.2.5 (*Langmead and Salzberg, 2012*). ChIP-seq peaks were called using the 'callpeak' function of MACS2.1.1 (*Zhang et al., 2008*). We used '−BAMPE' options of MACS2.1.1. To catalogue the binding sites over all the time points, we merged and reduced all peaks, and removed those falling within ±4 Mb of centromeric

heterochromatin regions. Duplicates were removed. Read numbers, Spearman's correlation coefficients between replicates, FRiP, and peak numbers are tabulated in *Supplementary file 7*.

## ATAC-seq data processing

Trimmed reads were aligned to GRCh38 genome assembly without mitochondrial chromosome (ChrM) using Bowtie1.0.0 (*Langmead et al., 2009*). Peaks were called using the 'callpeak' function, with '`-shift 37 -extsize 73`' options of MACS2.1.1. Peaks called with '`-broad`' and '`-narrow`' were combined and used for further analysis. Duplicates were removed.

## RNA-seq data analysis

For each time points, read count data were generated on RefSeq annotation using HTSeq0.6.1p1 (*Anders et al., 2015*). For *Figure 1C*, we used DESeq2 to obtain normalized count data and differentially expressed genes at each time point compared to any other time points with a threshold of | log2 fold change| >1 and false discovery rate <0.05. The resulting genes were clustered using k-means clustering. Principal component analysis was carried out using the built-in R function, prcomp(). PCA plot in *Figure 1D* was generated using ggplot2 (http://ggplot2.org). Loadings were computed, and top 1000 genes with high loadings were selected for component 1 and 2. Among 1000 genes, the ones that were not found in the other group were retained for further analysis. Heatmaps were generated from gene expression data with pheatmap (https://CRAN.R-project.org/package=pheatmap). Gene ontology (GO) analysis was performed with DAVID (*Huang et al., 2009*).

## TU annotation

TUs were annotated from TT-seq data as described (*Michel et al., 2017*), with modifications. Briefly, we used the GenoSTAN package in R/Bioconductor (*Zacher et al., 2017*) for segmenting the genome into two states, 'expressed' and 'not expressed', based on combined TT-seq data at 0 hr, 12 hr, 24 hr, and 96 hr. Posterior state probabilities were calculated, and the most likely state path (Viterbi) was obtained. TUs within 200 bp as well as TUs mapping to exons within GENCODE (Human, Release 25) annotated genes were merged together. The Jaccard Index was computed based on GENCODE (all support levels) and used as an expression cutoff.

## Classification of gene types

Gene types from GENCODE were taken if TUs overlapped with more than 60% of GENCODE annotation. Downstream RNAs (dsRNAs) were ncRNAs that lie within 1 kb downstream of protein-coding genes or products from stepwise termination of RNA polymerase II (*Schwalb et al., 2016*); ncRNAs that lie within 10 kb downstream of protein-coding genes with diminishing synthesis level over the genomic position. TUs within 1000 bp upstream and 500 bp downstream of the transcription start site (TSS) of protein-coding genes were assigned as upstream antisense RNA (uaRNA) if synthesized on upstream of the opposite strand as protein-coding gene TSS or convergent RNA (convRNA) if synthesized on the overlapping opposite strand as protein-coding gene TSS. Among rest of TUs, TUs overlapping with GENCODE (gene_type = gene) more than 20% of GENCODE were discarded. Of the remaining, ncRNAs within 1 kb of ATAC-seq and H3K4me1-ChIP-seq signals at any time point were assigned as eRNAs. The eRNAs within 1 kb of each other were merged regardless of synthesized strands, assuming to be synthesized from one enhancer. To compute length normalized counts of merged enhancers, counts were normalized to lengths of each eRNA TUs before merging. These normalized counts were added together for merged enhancers.

## Enhancer–promoter pairing

Each eRNA was paired with the putative target mRNAs by two methods; the neighboring method and 1 Mb method. For both methods, the eRNAs and mRNAs were taken from our annotation with TT-seq data and had to be synthesized at least at one time point. The neighboring method paired the eRNAs with mRNAs based on distances. First, intragenic eRNAs were paired with the mRNA in whose transcribed region they lay. The rest of the eRNAs was paired with both the closest upstream and downstream mRNAs. Of 12,900 pairs, 5786 pairs (45%) with differentially synthesized mRNAs were used (|log2FC| > 1, FDR < 0.05, computed by DESeq2). As a result, 4721 of 7624 enhancers were paired to 1790 genes (*Figure 2B*). The 1 Mb method paired the enhancers with the mRNAs

that were differentially synthesized (|log2FC| > 1, FDR < 0.05, computed by DESeq2), and within 1 Mb of the TSS of eRNAs. Pairs that had a Spearman's correlation coefficient between mRNA and the eRNA read counts higher than 0.4 were used. TAD pairing was carried out as for 1 Mb pairing, except that experimentally determined TADs (*Stik et al., 2020*) were used instead of the 1 Mb linear distance.

## TAD analysis

TAD information of BLaER cell line during transdifferentiation was obtained from *Stik et al., 2020*. We also confirmed that the enhancers and the paired target genes are within the same TAD, using previously published Hi-C data from the human monocytic leukemia cell line THP-1 (*Phanstiel et al., 2017*). We could use the data from the related cell line because TADs are known to be well conserved across cell types (*Rao et al., 2014*). Figures were generated using 'The 3D Genome Browser' (http://promoter.bx.psu.edu/hi-c/).

## Correlation between enhancer–promoter pairs

Enhancer–promoter (EP) pairs were classified based on the number of enhancers paired with the same promoter. We obtained the log2 fold change of length normalized mRNA and eRNA synthesis from 0 hr. For *Figure 4* and S4, we computed the Spearman's correlation coefficient using a sum of changes in eRNA synthesis as explanatory variable and a change in mRNA synthesis as target variable. The log2 fold change of eRNA and mRNA synthesis were regressed using the lm() function in R, and regression coefficient and coefficient of determination (R-squared, $R^2$) were obtained.

## Model to predict promoter activity based on the activity of multiple enhancers

Theoretical models were suggested by *Dukler et al., 2017*.

The additive relationship is described as follows:
Promoter activity = $\alpha \times$ (sum of enhancer activities) + $\beta$
The synergistic (exponential) relationship is described as:
Promoter activity = $\exp[\alpha \times$ (sum of enhancer activities) + $\beta]$
The logistic relationship is described as follows:
Promoter activity = $\delta/(1 + \exp[-(\alpha \times$ (sum of enhancer activities) + $\beta)])$

The constants $\alpha$, $\beta$, and $\delta$ describe the effect size of enhancer activities on their target promoter and vary between promoters. It can be assumed that these constants are fixed for each enhancer–promoter pair, even upon stimulation. The promoter activity was predicted from the sum of enhancer activities using generalized linear models. The additive and synergistic models were fitted with the use of normal error mode using the DEoptim package in R/Bioconductor and codes adapted from *Dukler et al., 2017*. The goodness of fit of the models was assessed using the BIC as in *Dukler et al., 2017*. Enhancer–promoter pairs with a good correlation of mRNA and eRNA synthesis (top 75% of Spearman's correlation coefficients) were further analyzed. First, we identified the genes with lowest BIC among each model, additive, synergistic, or logistic (*Figure 5—figure supplement 1D,E and G*). For the genes with the lowest BIC for additive or synergistic model (n = 523, 1017 with the neighboring or 1 Mb method, respectively), relative BIC was computed as a difference of BIC of the additive model compared to the exponential model. The synergistic regulatory mode was favored over the additive mode if the relative BIC was greater than 2. The regulatory mode was ambiguous if the relative BIC was between 0 and 2, and was considered to be additive if the relative BIC was less than 0. If BIC for the logistic model was the lowest (n = 250, 429 with the neighboring or 1 Mb method, respectively), we excluded the data point with the highest sum of enhancer activities and repeated from assessing BIC for each model.

## Superenhancer prediction using ROSE

Superenhancers are predicted as suggested in *Lovén et al., 2013*; *Whyte et al., 2013*. Briefly, ATAC-seq peaks, C/EBPα, BRD4, and H3K27Ac ChIP-seq peaks within 1 kb of our enhancers were stitched together if they fell within 12.5 kb. Signals were normalized to the peak widths. Highly ranked peaks were identified as superenhancers. The putative target genes of superenhancers are listed and compared with the genes predicted to be regulated by superenhancers in dbSUPER,

superenhancer database (*Khan and Zhang, 2016*; *Supplementary file 1*). Gene ontology analysis was performed on the putative target genes of superenhancers with DAVID (*Huang et al., 2009*; *Supplementary files 2–5*). Previously identified superenhancers from all available cell lines from dbSUPER, CD14+ cells, or CD19+ cells (*Hnisz et al., 2013*) were used to compare with the enhancers identified in this study. Specifically, acetylated H3K27 were used to classify superenhancers in CD14+ and CD19+ cells (*Hnisz et al., 2013*). Before the comparison, genome coordinates of our enhancers were converted to GRCh37/hg19 genome assembly (Human Genome Reference Consortium) using liftOver (*Hinrichs et al., 2006*).

### TF footprinting analysis

TF bound motifs were identified by PIQ, a TF footprinting method (*Sherwood et al., 2014*). First, GRCh38 genome was scanned with the motifs in Jaspar2018 CORE vertebrates database (*Khan et al., 2018*). With the combined coverage of two replicates of ATAC-seq, putative TF binding sites were computed at each time point, 0 hr, 12 hr, 24 hr, and 96 hr with a default threshold of PIQ. Significantly enriched motifs in late synergistic enhancers relative to additive enhancers were determined using Fisher's exact test (p-value < 0.05) for each time point. The SeqLogo function in the TFBSTools package (*Tan and Lenhard, 2016*) was used to generate motif logos. Using TFClass classification (*Wingender et al., 2018*), TFs in the same subfamily as enriched motifs were further analyzed.

## Acknowledgements

We thank the past and current members of the Cramer laboratory, in particular B Schwalb, M Lidschreiber, and S Sohrabi-Jahromi for help with bioinformatical analysis, K Maier and P Rus for sequencing and A Sawicka for advice on gene reporter assays. We also thank LAFUGA at the Gene Center, Ludwig-Maximilians-University of Munich, and TAL at University Medical Center Göttingen for sequencing. PC was supported by the Deutsche Forschungsgemeinschaft (SFB860, SPP1935, SPP2191), the European Research Council Advanced Investigator Grant TRANSREGULON (grant agreement No 693023), and the Volkswagen Foundation.

## Additional information

### Funding

| Funder | Grant reference number | Author |
|---|---|---|
| Max-Planck-Gesellschaft | Open-access funding | Patrick Cramer |
| Deutsche Forschungsgemeinschaft | SFB860 | Patrick Cramer |
| Deutsche Forschungsgemeinschaft | SPP1935 | Patrick Cramer |
| Deutsche Forschungsgemeinschaft | SPP2191 | Patrick Cramer |
| European Research Council | 693023 | Patrick Cramer |
| Volkswagen Foundation | | Patrick Cramer |

The funders had no role in study design, data collection and interpretation, or the decision to submit the work for publication.

### Author contributions

Jinmi Choi, Conceptualization, Data curation, Formal analysis, Validation, Investigation, Visualization, Methodology, Writing - original draft, Writing - review and editing, Designed research, performed all experiments except when stated otherwise, analysed and interpreted the data; Kseniia Lysakovskaia, Validation, Investigation, Methodology, Writing - review and editing, Set up and performed CRISPR validation and gene reporter assays; Gregoire Stik, Validation, Investigation, Methodology, Carried out CRISPR-based enhancer deletion and provided Hi-C data prior to publication; Carina

Demel, Data curation, Formal analysis, Carried out TU annotation; Johannes Söding, Conceptualization, Formal analysis, Supervision, Writing - review and editing, Advised on data analysis and manuscript writing; Tian V Tian, Methodology, Provided BLaER1 cell line and instructions for transdifferentiation; Thomas Graf, Conceptualization, Methodology, Writing - review and editing, Provided BLaER1 cell line and instructions for transdifferentiation, advised on manuscript writing; Patrick Cramer, Conceptualization, Resources, Supervision, Funding acquisition, Investigation, Visualization, Methodology, Writing - original draft, Project administration, Writing - review and editing

**Author ORCIDs**
Jinmi Choi https://orcid.org/0000-0002-4909-9473
Kseniia Lysakovskaia https://orcid.org/0000-0002-7829-4243
Gregoire Stik http://orcid.org/0000-0002-1404-1992
Johannes Söding http://orcid.org/0000-0001-9642-8244
Tian V Tian https://orcid.org/0000-0002-9906-0980
Thomas Graf http://orcid.org/0000-0003-2774-4117
Patrick Cramer https://orcid.org/0000-0001-5454-7755

**Decision letter and Author response**
Decision letter https://doi.org/10.7554/eLife.65381.sa1
Author response https://doi.org/10.7554/eLife.65381.sa2

# Additional files

## Supplementary files
• Supplementary file 1. List of superenhancers identified from chromatin accessibility, BRD4, C/EBPα, and H3K27Ac occupancies.

• Supplementary file 2. GO analysis on superenhancers identified from BRD4 signal.

• Supplementary file 3. GO analysis on superenhancers identified from H3K27Ac ChIP-seq signal.

• Supplementary file 4. GO analysis on superenhancers identified from C/EBPα signal.

• Supplementary file 5. GO analysis on superenhancers identified from chromatin accessibility signal.

• Supplementary file 6. Primer list.

• Supplementary file 7. Quality assessment of ChIP seq data sets.

• Transparent reporting form

## Data availability
RNA-seq, TT-seq, ChIP-seq, ATAC-seq data reported in this study were deposited with the National Center for Biotechnology Information Gene Expression Omnibus (accession number GSE131620). Hi-C data and H3K27Ac ChIP-seq in BLaER and Hi-C data in THP-1 cell lines that support the findings of this study are available with the National Center for Biotechnology Information Gene Expression Omnibus (accession GSE141226) and BioProject (accession PRJNA385337).

The following dataset was generated:

| Author(s) | Year | Dataset title | Dataset URL | Database and Identifier |
|---|---|---|---|---|
| Choi J, Lysakovskaia K, Stik G, Demel C, Soeding J, Tian TV, Graf T, Cramer P | 2020 | Evidence for additive and synergistic action of mammalian enhancers during cell fate determination | https://www.ncbi.nlm.nih.gov/geo/query/acc.cgi?acc=GSE131620 | NCBI Gene Expression Omnibus, GSE131620 |

The following previously published datasets were used:

| Author(s) | Year | Dataset title | Dataset URL | Database and Identifier |
|---|---|---|---|---|
| Stik G, Casadesus MV, Graf T | 2020 | CTCF is dispensable for cell fate conversion but facilitates acute cellular responses [Hi-C] | https://www.ncbi.nlm.nih.gov/geo/query/acc.cgi?acc=GSE141226 | NCBI Gene Expression Omnibus, GSE141226 |
| UNC Chapel Hill | 2017 | in situ Hi-C data of THP-1 cells untreated and treated with PMA | https://www.ncbi.nlm.nih.gov/bioproject/?term=PRJNA385337 | NCBI BioProject, PRJNA385337 |

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
