## [Decision Letter]

**Acceptance summary:**

The authors provide insight into genome control mechanisms that are used to generate context-specific patterns of gene activity that underlie the distinct functions of cells and tissues. The study's novelty lies in the application of eRNA kinetics during differentiation and the finding that synergistic enhancer/promoter sets are enriched for both lineage-determining genes and binding of cell type-specifying transcription factors.

**Decision letter after peer review:**

[Editors’ note: the authors submitted for reconsideration following the decision after peer review. What follows is the decision letter after the first round of review.]

Thank you for submitting your work entitled "Enhancer synergy can drive human cell fate determination" for consideration by *eLife*. Your article has been reviewed by three peer reviewers, and the evaluation has been overseen by a Reviewing Editor and a Senior Editor. The following individual involved in review of your submission has agreed to reveal their identity: Harinder Singh (Reviewer #2).

Our decision has been reached after extensive consultation among the reviewers. Based on these discussions and the individual reviews below, we regret to inform you that your work will not be considered further for publication in *eLife*.

The reviewers appreciated the study's attempt at addressing several important and timely questions regarding the developmental control of mammalian gene transcription. In particular, they were supportive of the rich data sets, the kinetic analyses and the use of TT-seq to analyze eRNAs (as a proxy of endogenous enhancer activity) and promoter output. Nevertheless, the three reviewers expressed a variety of concerns (see below) which, following extensive discussion, led to a decision to decline your submission. From reading the reviewers' comments you will see that a variety of further experiments would have been necessary to satisfy them that some conclusions are fully warranted. When coming to this decision, we and the reviewers took into consideration the likely time required to perform these experiments. Reviewers expressed an interest in seeing a new submission containing additional data that support (or otherwise) your initial manuscript's predictions, should you choose to submit this to *eLife*.

Reviewer #1:

The manuscript tackles an interesting question, but it completely lacks experimental verification of the hypothesis generated by the very intricate but poorly explained analysis. Without experimental validation of the prediction that some enhancer sets act synergistically while other are only additive, the major conclusions of the manuscript remain preliminary.

1) The manuscript is devoid of performance metrics for the sequencing experiments, making it impossible to gauge data quality. No comparison/reproducibility analysis of the replicates is presented, nor data on IP quality, peak numbers, read counts, IP efficiency. The super-enhancer analysis is not internally validated by assessing whether the super-enhancers identified using BRD4, C/EBPa or ATAC-seq make biological sense (are the identified super-enhancers lineage genes?), which would be important for the conclusion that synergistic enhancers are distinct from super-enhancers.

2) For validation, enhancer cooperativity could be tested in reporter assays with a handful of enhancers of either type, alone or in combination. Alternatively, Cas9-mediated knockout of increasing numbers of enhancers in a given gene/enhancer set would be predicted to have linear or non-linear effects on their target gene, depending on whether they are synergistic or additive.

3) The use of estrogen to activate the fusion protein could lead to estrogen receptor-dependent enhancer activation. The authors should clarify whether they see these effects, and if so, how they discern them from the C/EBPa-mediated effects.

4) The Discussion lacks discussion of previous findings; for example, inducible TF binding has previously been shown induce histone methylation (H3K4me1) at enhancers and nucleosome remodeling (doi:10.1186/s13059-016-0897-0, doi:10.1016/j.molcel.2010.05.004), which is corroborated by the current manuscript. For short-term transcriptional responses, eRNA transcription elongation has been shown to be required for MLL4-mediated H3K4me1/2 deposition at enhancers, while in this manuscript eRNA transcription occurs after H3K4me1 deposition – how do the authors explain this mechanistic difference?

Reviewer #2:

The authors are attempting to address several fundamental issues concerning the developmental control of mammalian gene transcription by deploying various genome wide methodologies and computationally integrating their findings. They have previously described a useful model system to analyze such regulatory processes. The experimental system involves a B cell line that undergoes transdifferentiation into a macrophage by the ectopic and inducible expression of the transcription factor C/EBPa. Using this system and performing transient transcriptome sequencing (TT-seq), to analyze eRNAs and promoter output, as well as C/EBPa ChIP-seq, H3K4me1 ChIP-seq and ATAC-seq they are able to put together a temporal sequence of molecular events for C/EBPa targeted genes. The experiments are rigorous and enhanced by the kinetic analyses.

Such temporal analysis has been performed in other systems and several of the key conclusions reached are similar to those of earlier studies. In particular that C/EBPa binds to a class of genomic sites which are accessible as well as those that are inaccessible (ATAC-seq). For the latter set, chromatin accessibility follows C/EBPa binding and H3K4me1.

The new element in this work is the genome-wide analysis of eRNAs, used as a proxy for endogenous enhancer activity, and their correlation with the activities of nearest (or within I Mb) promoters. Based on such analysis the authors conclude that for promoters that are acted on by multiple enhancers, most enhancers function additively. However for a smaller set of genes, enhancers appear to function synergistically and these genes are enriched for cell type specific functions.

A current model for super enhancer action involving liquid-liquid phase transition (molecular condensate formation with TFs, chromatin modifiers and the RNA Pol II machinery) is suggested to underlie the basis of enhancer synergy.

There are two major issues to be addressed:

1) In this study, enhancers are inferred to be connected to their cognate promoters on the basis of nearest neighbors or by using an arbitrary genomic distance (I Mb). Although such criteria are widely used they have been supplanted by Hi-C or promote capture Hi-C analysis as a means of directly analyzing enhancer-promoter interactions. Given that this connectivity is critical for the subsequent analysis and conclusions focusing on enhancer action (additive versus synergistic) on different types of target genes, the authors need to use of one of the above approaches.

2) Although not emphasized in the Abstract the authors fail to uncover evidence for super enhancers (clusters of enhancers spanning several kb) operating in their cellular system in the selective control of key differentiation genes. They suggest that their method of utilizing eRNAs rather than mediator binding revealed by ChIP-seq is better at distinguishing functionally distinct classes of enhancers. This major conclusion needs to be elaborated and deserves to be featured in the Abstract. From a mechanistic standpoint there appears to be a problem that needs to be better discussed by the authors. As noted above a current model for super enhancer action involving liquid-liquid phase transition is invoked by the authors to underlie the basis of enhancer synergy. However in their study the identified super enhancers appear to function either additively or synergistically. Can these different types of analyses be reconciled or perhaps not?

Reviewer #3:

In this manuscript, Choi and colleagues study enhancer cooperativity – i.e. how multiple enhancers collaboratively regulate the transcription of their target genes – during transdifferentiation of human B cells to macrophages at a genome-wide scale. To do so, they used an estrogen-inducible system driven by the C/EBPα pioneer factor and monitored transcriptional and chromatin dynamics over time. By using RNA-Seq, they first show that two main waves of transcriptional changes underline B cell to macrophage transdifferentiation. Then, they combined Transient Transcriptome Sequencing (TT-Seq) with newly generated ATAC-Seq and ChIP-seq for the H3K4me1 enhancer mark to identify enhancer elements that are transcribed in at least one of the timepoints, highlighting a subset of enhancers whose transcription changes upon transdifferentiation. To further disentangle the order of events underpinning transdifferentiation they performed ChIP-seq for the C/EBPα TF, whose binding is associated with chromatin opening and later transcriptional activation, further supporting the proposed role of C/EBPα as a pioneering factor.

In order to tackle enhancer cooperative behaviors, the authors used two distinct assignment strategies to pair enhancers to their target promoter(s) and report that (1) mRNA transcription of target genes correlate with the number of assigned enhancers and that (2) the sum of eRNA transcription at all assigned enhancers outperforms eRNA transcription at individual loci in predicting mRNA transcription at multiple enhancer genes. By fitting additive, synergistic and logistic models to their time-course datasets, the authors find enhancers predicted to cooperate in an additive but also synergistic fashion, meaning that mRNA output of their assigned target gene(s) is higher than expected from the sum of eRNA transcription at cognate enhancers. Finally, they show that synergistic enhancers are mostly associated with cell-type specific genes and are enriched for cell-type specific TF binding motifs, although they do not show higher C/EBPα, chromatin accessibility or BRD4 levels, as previously reported for cell-type specific "super-enhancer" hubs.

The authors address an important and timely question in the field of gene regulation, i.e. how do multiple enhancers cooperate to regulate their target genes. To achieve this goal, the authors produce a compelling and rich dataset, analyze various aspects of it and apply a modeling approach to assign enhancers to additive and synergistic modes.

Overall, this study presents a comprehensive characterization of transcription and various chromatin aspects (chromatin accessibility, H3K4me1, factor binding) for a transdifferentiation timecourse from human B lineage cells to macrophages – a true tour-de-force. We however find that the authors partly overstate their findings and draw overly strong conclusions without adequately discussing caveats. We therefore recommend that the authors carefully revise their manuscript prior to publication to address two major concerns. The two major concerns relate to 1. some important untested assumptions that underlie the key finding of the manuscript, i.e. enhancer synergy and 2. partly overly strong conclusions that for example incorrectly imply causation from correlative analyses.

1) Potential caveats to the key finding – enhancer synergy

The additive and synergistic cooperative behaviors reported by the authors heavily rely on (i) eRNAs – or TT-seq signals – accurately reflecting enhancer activity, (ii) a truly comprehensive set of enhancers, and (iii) enhancer-promoter assignment strategies. The authors need to address or discuss these points in more depth, as their key finding, i.e. the existence of enhancer synergy, crucially relies on all three assumptions being correct – any underestimation of the number of enhancers or their activities will unavoidably make the enhancers appear synergistic even if they are not. We comment on each of these points separately and then discuss some implications in more depth below.

(i) The authors treat this caveat rather superficially, referring to two recent publications (Henriques et al., 2018 and Mikhaylichenko et al., 2018) that study transcription at enhancers in *Drosophila*. Whether or not transcription rates truly reflect enhancer activities is far from clear: Mikhaylichenko et al. for example state "the levels and directionality of transcription are highly varied among active enhancers. […] enhancer RNA (eRNA) production and activity are not always strictly coupled", and in human HeLa and HepG2 cells, CAGE-eRNA-signals do not quantitatively correlate with enhancer activities (Andersson et al., Nature 2014; see supplementary figures 9a and 10a). In addition (more minor), at the core of TT-Seq is the transient sparse labeling of newly synthetized RNAs and their subsequent enrichment by affinity purification. We recommend discussing in more detail TT-seq's sensitivity and ability to reliably detect short, unstable and lowly abundant eRNAs (compared to more stable and longer mRNA molecules).

ii) The authors define enhancers as TT-Seq Transcription Units (TUs) and further require them to be enriched for both H3K4me1 and ATAC-Seq signals. Through these latter criteria, only 8,165 of 26,130 TUs are defined as enhancers, leaving the vast majority of about 18,000 TUs unassigned. While stringent filtering makes much sense in general, a systematic underestimation of the number of enhancers might strongly confound the analysis of enhancer cooperativity. As the H3K4me1 mark is not required for enhancer activity (Dorighi et al., 2017; Rickels et al., Nature Genetics, 2017), the authors should consider whether these unannotated TUs might also stem from enhancer elements and e.g. from gene loci that show synergistic effects.

iii) In addition to potentially missing enhancers by an overly stringent enhancer definition (point ii), the number of enhancers per gene can also be underestimated at the step of enhancer-to-gene assignment. To mitigate this problem, the authors use two complementary assignment strategies, "neighboring" and "1Mb", which make sense yet have their caveats. While the 1Mb method should be more inclusive, it for example imposes a 0.4 correlation coefficient between eRNA and mRNA changes, thus limiting the pertinence of this approach for further hypothesis testing (see major concern 2). More importantly though, the two approaches don't seem to yield very consistent results: 136 genes (92+44; subsection “Enhancer cooperation can be additive or synergistic”) were classified as synergistic by the neighboring method and 194 by the 1Mb method (subsection “Enhancer synergy is a robust phenomenon”), yet only 57 could be tested by both methods of which 33 were consistently classified as synergistic. While the authors claim that this corresponds to a strong overlap (a Fisher exact p-value is provided with 3.5e-11), it is unclear why so few genes can be tested by both methods and why of the 136 or 194 synergistically regulated genes, respectively, only 33 are consistent between methods. At the very least, these numbers suggest that the classification to the additive or synergistic models strongly depends on the enhancer-to-gene assignment.

With these caveats in mind, it is not clear whether the manuscript conclusively demonstrates the existence of synergistic enhancer cooperativity and what to recommend. A systematical under-estimation of enhancer activities or their dynamics (Figure 3B indicates that eRNAs are only detected rather late) would bias the analysis towards seemingly synergistic enhancer cooperativity, especially for genes that are highly expressed and/or strongly upregulated.

To reach strong conclusions, one would probably have to directly test combinations of enhancers or perform CRISPR-Cas9-mediated deletion of additive and synergistic enhancers with similar individual activities (as has been done in papers claiming or refuting enhancer synergy of e.g. superenhancers). One would expect that the deletion of an additive enhancer would have a limited impacted on downstream mRNA transcription and transdifferentiation efficiency, while the deletion of a synergistic enhancer would lead to stronger effects and might potentially impede transdifferentation.

However, especially given the data-richness of the manuscript, such major additional efforts seem to be beyond the scope of this project. Maybe the authors could more openly discuss the caveats listed above and how they might confound the analyses and/or perform additional sanity checks, maybe some of the following:

– simulations to assess the potential influence of underestimating enhancer activities or enhancer number on the additivity vs. synergy estimates. For example, the authors could determine how a simulated increase of eRNA transcription (e.g. by 5%, 10%, 20% or 50%) impacts the assignment of gene loci to additive, synergistic and/or logistic classes.

– a systematic analysis of potential correlations of synergistic genes with the number of enhancers. The author used gene loci containing 2 to 20 enhancers for model fitting and it would be informative whether synergistic loci overall contain more enhancers compared to additive ones.

2) Data presentation and overly strong conclusions

The authors describe and analyze their datasets by a combination of rule-based filtering and direct comparisons. This strategy is valid, yet can easily suffer from circular logic if the rules influence the comparisons, which can render results trivial. We recommend that the authors carefully revise their manuscript to remove analyses that might suffer from circular logic or discuss such caveats and tone down their conclusions.

For example, the authors require "nucleosome depletion" to define enhancers and subsequently study 4550 C/EBPa binding sites "with low or undetectable chromatin accessibility at 0h", which unavoidably leads to the observation that chromatin opens. Similarly, the "1Mb" method to assign enhancers to promoters includes a requirement that the activities need to be correlated, which means that correlations (e.g. those shown in Figure 4—figure supplement 1 E-F) cannot be seen as results.

Similarly, the authors draw strong causal conclusions from correlative analyses (e.g. "Chromatin opening enables enhancer RNA synthesis”, or "Enhancer synergy can drive cell fate determination", title), which should be avoided.

[Editors’ note: further revisions were suggested prior to acceptance, as described below.]

Thank you for submitting your article "Additive and synergistic action of mammalian enhancers during cell fate determination" for consideration by *eLife*. Your article has been reviewed by three peer reviewers, and the evaluation has been overseen by a Reviewing Editor and Kevin Struhl as the Senior Editor. The following individual involved in review of your submission has agreed to reveal their identity: Harinder Singh (Reviewer #1).

The reviewers have discussed the reviews with one another and the Reviewing Editor has drafted this decision to help you prepare a revised submission.

Summary:

Choi et al. analyze the transcription changes at enhancers and promoters during estrogen-induced transdifferentiation of a human CEBPa-ER-expressing B cell lymphoma cell line into a macrophage state. By integrating diverse data (C/EBPa binding pattern, H3K4me1 ChIP-seq, ATAC-seq and TT-seq) at enhancers, the authors propose that C/EBPa binding opens chromatin, increases H3K4me1 in its vicinity, and is followed by eRNA transcription. They conclude that C/EBPa functions as a pioneer factor. The manuscript studies gene transcription, enhancer activities and chromatin properties during transdifferentiation of human B cells to macrophages. While the sums of enhancer activities explain gene transcription for most genes, they do not for over 130 gene loci, suggesting that the enhancers in these loci might cooperate synergistically. The study's novelty lies in the application of eRNA kinetics during differentiation and the finding that synergistic enhancer/promoter sets are enriched for both lineage-determining genes and binding of cell type-specifying transcription factors.

The authors define temporally co-regulated pairs and cohorts of enhancers and promoters, and find 3-fold more mRNA/eRNA sets that fit an additive abundance model over those that exhibit an exponential (synergistic) relationship. Although the synergistic sets of enhancers are enriched for cell type-specific genes, similar to what has been described for super-enhancers, they appear to be distinct from super-enhancers defined previously in the same cells. Motif analysis of deeply sequenced ATAC-seq data indicates that synergistic enhancer sets are enriched of additive enhancer sets for cell type-specific transcription factors involved in lineage determination.

The authors have substantially revised their manuscript in response to previous comments. In the light that experimental verification of the predicted additive versus synergistic enhancer action was not feasible, the message of the manuscript could be strengthened further in multiple ways. First, the authors should disclose lists of (i) identified additive and synergistic enhancer cohorts and associated genes and (ii) the super-enhancers they identified, including their genomic coordinates. Similarly, the authors should disclose the full results of the GO analysis for all enhancer/super-enhancer set. Additionally, they could provide genome browser tracks and/or more screenshots of example loci. Second, the authors could verify enrichment of lineage-determining transcription factors in synergistic enhancers by ChIP-seq. Finally, the manuscript would benefit from focusing further on the novel aspects it presents and attenuating remaining claims that could be perceived as overstatements.

Revisions:

1) As the authors could not directly validate enhancer synergy, they indirectly infer its existence by the following logic: They show that the sum of enhancer activities does not match the transcription levels for some genes but leaves some gap. They use this gap to argue that the identified enhancers must be super-additive, i.e. synergistic, in order to fill this gap. While plausible, such an inference is different from the unambiguous demonstration of enhancer synergy. This fact will need to be better reflected in the Title and Abstract. Otherwise, a superficial reader might assume that synergy has been demonstrated directly. This wording could for example directly state the authors' reasoning, namely that a gap in observable enhancer activities suggests the existence of enhancer synergy for some genes. The last sentence of the revised Abstract needs to provide clarity on the comparison being made between synergistically functioning enhancer clusters and super-enhancers. It could read: "Enhancer synergy appears to be dependent on binding of cell type-specific transcription factors and such interacting enhancers are not predicted from occupancy or accessibility data that are used to detect super-enhancers."

2) The claim that enhancer synergy "could not be predicted from occupancy or accessibility data that are used to detect super-enhancers" hinges on the comparison of eRNA-derived enhancer cohorts with super-enhancers. This comparison retains two issues that still need to be addressed. First, super-enhancers represent regions of high enhancer activity, originally defined by Med1 or H3K27ac ChIP-seq. Consequently, open chromatin (ATAC-seq) or the locations of a pioneer factor (C/EBPa) that induces open and poised – but not necessarily active – chromatin is not necessarily expected to yield meaningful results in terms of defining genomic regions with cell type-specific and lineage-defining enhancer and gene activity. This would leave the BRD4 ChIP-seq data as the sole signal to define meaningful super-enhancers. However, its high peak count of >98,000 raises concerns about potential ChIP quality issues that unfortunately cannot be assessed because the authors fail to provide additional information on these ChIPs other than the peak count. That the Spearman rho values for BRD4 ChIP replicates fluctuate between 0.72 and 0.99 also do not inspire confidence. Here, a H3K27ac as a broader and more robust ChIP-seq marker of enhancer and transcription activity might help to improve super-enhancer calling, should BRD4 turn out to be sub-optimal. Second, save for a handful of super-enhancer-associated genes that have been included in the revised manuscript, the biological relevance (i.e. gene ontology term enrichment) of the identified super-enhancer-associated genes cannot be verified by readers, since the authors provide neither the corresponding list of genes nor an analysis of the super-enhancer-enriched GO terms. At a minimum, the authors should provide these gene lists for the different super-enhancer methods and additional quality information of the ChIP-seqs used to identify super-enhancers, perform GO term analyses and provide the full list of significantly enriched GO terms, to be able to gauge whether they identify true super-enhancers with the corresponding expected characteristics (cell type specificity, enrichment near lineage-determining genes and transcription factors), and whether these are indeed distinct from synergistic enhancers.

3) The analysis of transcription factors in synergistic enhancers versus additive ones is not fully developed and contains inaccuracies and confusing interpretations of the data. Specifically, there is confusion over the two motifs that were determined to be enriched in additive enhancers, for FOS and E2F family factors.

a) The motif for "FOS family" factors (TGAc/gTCA) is actually the motif to which most AP-1 family heterodimers (Fonseca, Nat Comm 2019). Indeed, AP-1 family factors (and not just FOS) play cell type-specific roles in macrophage development and function and their activity is low or undetectable at B cell enhancers by motif analysis (Heinz et al., 2010). Consequently, the statement that "additive enhancers were not [enriched for macrophage-specific TFs]" is incorrect, especially in the context of this study of B cell (-like cell) trans-differentiation into macrophage-like cells, and it is not questionable whether FOSL1 as given in Figure 6—figure supplement 1C is indeed the factor that binds these motifs, and which of the Jun-like factors that become upregulated are involved in AP-1 heterodimer binding to them.

b) There appears to be a disconnect between the interpretation of the authors' finding that E2F motifs are enriched in additive enhancers, their statement that E2F factors have regulatory functions in macrophages and the expression data in Figure 6B showing that E2F factors are rapidly downregulated upon estrogen addition, which is in line with the cell cycle arrest observed in these cells when leaving the B cell-like state (Rapino et al., 2013). Together, this indicates that E2F factors would only be able to bind to additive enhancers and play a role in their activation in the proliferative B cell-like state, but not in the macrophage-like state. If so, then why are E2F motifs enriched in late-time point enhancers?

4) The number of C/EBPa ChIP-seq peaks is surprisingly low for an overexpressed pioneer transcription factor, especially while other ChIPs in the data set have unusually large numbers of peaks (>98,000 peaks for BRD4 seems very high for a non-DNA-binding coregulator). These peak numbers are unusual and raise data quality concerns, and for C/EBPa might also explain the small number of C/EBPa-defined "super-enhancers". Further, the peak numbers identified by ATAC-seq are not provided, and here the Spearman's rho was only calculated for promoters, which is problematic in a manuscript dedicated to enhancers and raises additional data QC concerns. Providing additional QC data such as FRiP and missing peak numbers, preferentially in a tabulated form, as well as additional genome browser track screenshots at different resolutions (also of the BRD4 ChIP-seqs) are recommended.

---

## [Author Response]

[Editors’ note: the authors resubmitted a revised version of the paper for consideration. What follows is the authors’ response to the first round of review.]

Reviewer #1:The manuscript tackles an interesting question, but it completely lacks experimental verification of the hypothesis generated by the very intricate but poorly explained analysis. Without experimental validation of the prediction that some enhancer sets act synergistically while other are only additive, the major conclusions of the manuscript remain preliminary.1) The manuscript is devoid of performance metrics for the sequencing experiments, making it impossible to gauge data quality. No comparison/reproducibility analysis of the replicates is presented, nor data on IP quality, peak numbers, read counts, IP efficiency.

We have added these metrics to the Materials and methods section. The average read numbers obtained for total RNA-seq were 1x10^7^, and 2x10^8^ for TT-seq. Spearman's correlation coefficients between replicates were 0.99 for total RNA-seq and TT-seq on protein coding genes at all time points. The average read numbers obtained were 4x10^7^ for C/EBPa ChIP-seq, 7x10^7^ for H3K4me1 ChIP-seq and 2x10^7^ for BRD4 ChIP-seq. Spearman's correlation coefficients between replicates were 0.94-0.98 for C/EBPα ChIP-seq, 0.96-1 for H3K4me1 ChIP-seq, and 0.72-0.99 for BRD4 ChIP-seq. The number of peaks were 14,561, 55,904 and 98,757 for C/EBPa, H3K4me1 and BRD4, respectively. The average read numbers obtained were 3x10^7^ for ATAC-seq. Spearman's correlation coefficients between replicates were 0.93-0.98 for ATAC-seq on promoters.

The super-enhancer analysis is not internally validated by assessing whether the super-enhancers identified using BRD4, C/EBPa or ATAC-seq make biological sense (are the identified super-enhancers lineage genes?), which would be important for the conclusion that synergistic enhancers are distinct from super-enhancers.

Superenhancers are identified according to the common procedure as enhancers that show high signals of BRD4 ChIP-seq, C/EBPa ChIP-seq or ATAC-seq. As previously reported, some of these enhancers target cell type-specific genes, such as CD14, DDIT4, JUNB, FOSL2 etc. However, there are some superenhancers commonly found at different cell states, for example at DNMT1, JMJD8 and POLE genes. As shown in Figure 5—figure supplement 3B, some of the superenhancers identified from macrophage-like cells and their target genes are transcribed more in B-cells (0h) than macrophage-like cells (96h). Altogether, the superenhancers identified using BRD4, C/EBPa ChIP-seq and ATAC-seq include both lineage genes and common genes.

2) For validation, enhancer cooperativity could be tested in reporter assays with a handful of enhancers of either type, alone or in combination. Alternatively, Cas9-mediated knockout of increasing numbers of enhancers in a given gene/enhancer set would be predicted to have linear or non-linear effects on their target gene, depending on whether they are synergistic or additive.

We have spent one year to conduct the suggested experiments, unfortunately without success. We have now decided to submit the revised version without the suggested experiments. We have changed the text and conclusions to reflect the lack of validation by enhancer knockouts/reporter gene assays. Our attempts are summarized below as an explanation to the reviewer. We note we are not able to continue these efforts.

a) CRISPR/Cas9 enhancer knockout experiments:

We first tried to obtain stable cell lines containing Cas9-mediated knockouts of increasing numbers of enhancers but this turned out to be impossible. Sometimes heterozygous cell lines resulted, and in other cases we could not obtain the desired knockouts. We believe that the enhancers we target are critical for cell growth, preventing analysis by CRISPR knockout.

b) Reporter gene assays

Due to these difficulties, we switched to reporter gene assays as suggested by the reviewer. For the luciferase reporter assay, we selected two representative gene candidates with three additive or synergistic enhancers (FTH1 and PELI1 respectively). We cloned target gene promoters alone or in different combinations with one, two or three enhancers (5 constructs per each candidate) into luciferase gene containing plasmid. We performed transdifferentiation of BLaER cells until 96 h and collected induced macrophage-like cells (iMacs). We transfected iMacs with different constructs and measured luciferase activity after 24 h of transfection using Dual-Luciferase Reporter Assay System (Promega, E1910). In total, we tested two transfection systems: electroporation with Neon Transfection System (Thermo Fisher Scientific) and lipofection with FuGENE HD Transfection Reagent (Promega, E2311). Unfortunately, it was not possible to get reproducible results for the experimental constructs from the technical and biological replicates due to the complexity of the system. Another complication stems from the need to use (as we did) the endogenous target promoter. Please understand we have to investigate reporter gene activity under transdifferentiation conditions and to our knowledge this has never been achieved. Provided the fluctuations often observed with reporter gene assays, in retrospect it is not surprising that we could not obtain reproducible signals.

3) The use of estrogen to activate the fusion protein could lead to estrogen receptor-dependent enhancer activation. The authors should clarify whether they see these effects, and if so, how they discern them from the C/EBPa-mediated effects.

We focus our enhancer activation analysis on C/EBPa binding sites (Figure 3). Additionally, the enhancer cooperation we observe would not be a consequence of C/EBPa activation only, but rather a general behavior of enhancers that comes from the cooperative actions of TFs (as shown in Figure 6). Thus, estrogen receptor dependent enhancer activation, if any, would not change our conclusion.

In addition, a corresponding inducible mouse cell (C10) is available with the exogenous estrogen receptor fused to C/EBPa (Rapino et al., 2013). The effect of estrogen receptor activation could be tested by comparing the gene expression between an inducible cell line (C10) and the mother cell line (HAFTL) treated with b-estrogen treatment. As shown in a MA (ratio intensity) plot (GSE17316), no significant changes in gene expression were observed (adjusted p.val < 0.05). (Figure 1—figure supplement 1C). This is now mentioned in the text.

4) The Discussion lacks discussion of previous findings; for example, inducible TF binding has previously been shown induce histone methylation (H3K4me1) at enhancers and nucleosome remodeling (doi:10.1186/s13059-016-0897-0, doi:10.1016/j.molcel.2010.05.004), which is corroborated by the current manuscript.

We thank the reviewer for pointing this out and we added the missing references and included these points in discussion. It is well agreed that TF binding induces histone methylation (H3K4me1) and nucleosome remodeling at enhancers. However, whether eRNA transcription is required for H3K4me1 deposition at enhancers is still controversial. The study mentioned in the reviewer's comment suggests that inhibition of Pol II elongation reduces H3K4me2 (Kaikkonen et al., 2013).

For short-term transcriptional responses, eRNA transcription elongation has been shown to be required for MLL4-mediated H3K4me1/2 deposition at enhancers, while in this manuscript eRNA transcription occurs after H3K4me1 deposition – how do the authors explain this mechanistic difference?

We thank the reviewer for pointing this out. It has also been shown that knockout of MLL3/4 causes reduction of enhancer Pol II occupancy and eRNA synthesis (Dorighi et al., 2017). Moreover, H3K4me1 can precede nucleosomal depletion, which is necessary step prior to transcription (current study, Jubb, Boyle, Hume, and Bickmore, 2017, Creyghton et al., 2011; Rada-Iglesias et al., 2011; Zentner et al., 2011; Bogdanovic et al., 2012; Bonn et al., 2012; Mercer et al., 2011; Rada-Iglesias et al., 2012; Wamstad et al., 2012). Collectively, H3K4me1 and enhancer transcription likely affect each other. We edited the manuscript accordingly.

Reviewer #2:The authors are attempting to address several fundamental issues concerning the developmental control of mammalian gene transcription by deploying various genome wide methodologies and computationally integrating their findings. They have previously described a useful model system to analyze such regulatory processes. The experimental system involves a B cell line that undergoes transdifferentiation into a macrophage by the ectopic and inducible expression of the transcription factor C/EBPa. Using this system and performing transient transcriptome sequencing (TT-seq), to analyze eRNAs and promoter output, as well as C/EBPa ChIP-seq, H3K4me1 ChIP-seq and ATAC-seq they are able to put together a temporal sequence of molecular events for C/EBPa targeted genes. The experiments are rigorous and enhanced by the kinetic analyses.Such temporal analysis has been performed in other systems and several of the key conclusions reached are similar to those of earlier studies. In particular that C/EBPa binds to a class of genomic sites which are accessible as well as those that are inaccessible (ATAC-seq). For the latter set, chromatin accessibility follows C/EBPa binding and H3K4me1.The new element in this work is the genome-wide analysis of eRNAs, used as a proxy for endogenous enhancer activity, and their correlation with the activities of nearest (or within I Mb) promoters. Based on such analysis the authors conclude that for promoters that are acted on by multiple enhancers, most enhancers function additively. However for a smaller set of genes, enhancers appear to function synergistically and these genes are enriched for cell type specific functions.A current model for super enhancer action involving liquid-liquid phase transition (molecular condensate formation with TFs, chromatin modifiers and the RNA Pol II machinery) is suggested to underlie the basis of enhancer synergy.There are two major issues to be addressed:1) In this study, enhancers are inferred to be connected to their cognate promoters on the basis of nearest neighbors or by using an arbitrary genomic distance (I Mb). Although such criteria are widely used they have been supplanted by Hi-C or promote capture Hi-C analysis as a means of directly analyzing enhancer-promoter interactions. Given that this connectivity is critical for the subsequent analysis and conclusions focusing on enhancer action (additive versus synergistic) on different types of target genes, the authors need to use of one of the above approaches.

Although we used two criteria that are widely accepted, we followed the reviewer’s request and now also incorporated Hi-C data to confirm the enhancer-promoter interactions. TAD boundaries were computed from the Hi-C data and used to study whether the enhancer promoter pairs (EP pairs) are located within the same TAD. Confirming our analysis, 98.3% of the EP pairs obtained by the neighboring method were found within the same TAD, and 96.2% of the EP pairs obtained with 1Mb method were found within the same TAD at one or more time points. We also employed the third pairing method by pairing enhancers to the correlating promoters within the same TAD obtained from Hi-C data. The synergistically regulated genes found with the neighboring and TAD pairing methods significantly overlapped, supporting our conclusions on synergistic enhancer action. This information was added to the text and figures.

2) Although not emphasized in the Abstract the authors fail to uncover evidence for super enhancers (clusters of enhancers spanning several kb) operating in their cellular system in the selective control of key differentiation genes. They suggest that their method of utilizing eRNAs rather than mediator binding revealed by ChIP-seq is better at distinguishing functionally distinct classes of enhancers. This major conclusion needs to be elaborated and deserves to be featured in the Abstract. From a mechanistic standpoint there appears to be a problem that needs to be better discussed by the authors. As noted above a current model for super enhancer action involving liquid-liquid phase transition is invoked by the authors to underlie the basis of enhancer synergy. However in their study the identified super enhancers appear to function either additively or synergistically. Can these different types of analyses be reconciled or perhaps not?

We agree this is an important finding. As suggested by the reviewer, we have now mentioned this point in the Abstract and discuss it better in the manuscript, but avoiding speculative conclusions. Briefly, superenhancers are identified as enhancers with high signals of BRD4 ChIP-seq, C/EBPa ChIP-seq or ATAC-seq. As previously reported, some of these enhancers target cell type specific genes, such as CD14, DDIT4, JUNB, FOSL2 etc. However, there are some superenhancers commonly found at different cell states, for example at DNMT1, JMJD8 and POLE genes. As shown in Figure 5—figure supplement 3B, some of the superenhancers identified from macrophage-like cells and their target genes are transcribed more in B-cells (0h) than macrophage-like cells (96h). Altogether, the superenhancers identified using BRD4, C/EBPa ChIP-seq and ATAC-seq include both lineage genes and common genes. It has been shown that not all super enhancers act synergistically (see Introduction). Meanwhile, the synergistic enhancers identified with RNA synthesis clearly showed higher frequency of long-range EP interactions than additive enhancers (see above). A recent report also showed that BRD4 inhibition does not disrupt EP interactions (biorxiv: doi: https://doi.org/10.1101/848325), and this was included in the Discussion.

Reviewer #3:In this manuscript, Choi and colleagues study enhancer cooperativity – i.e. how multiple enhancers collaboratively regulate the transcription of their target genes – during transdifferentiation of human B cells to macrophages at a genome-wide scale. To do so, they used an estrogen-inducible system driven by the C/EBPα pioneer factor and monitored transcriptional and chromatin dynamics over time. By using RNA-Seq, they first show that two main waves of transcriptional changes underline B cell to macrophage transdifferentiation. Then, they combined Transient Transcriptome Sequencing (TT-Seq) with newly generated ATAC-Seq and ChIP-seq for the H3K4me1 enhancer mark to identify enhancer elements that are transcribed in at least one of the timepoints, highlighting a subset of enhancers whose transcription changes upon transdifferentiation. To further disentangle the order of events underpinning transdifferentiation they performed ChIP-seq for the C/EBPα TF, whose binding is associated with chromatin opening and later transcriptional activation, further supporting the proposed role of C/EBPα as a pioneering factor.In order to tackle enhancer cooperative behaviors, the authors used two distinct assignment strategies to pair enhancers to their target promoter(s) and report that (1) mRNA transcription of target genes correlate with the number of assigned enhancers and that (2) the sum of eRNA transcription at all assigned enhancers outperforms eRNA transcription at individual loci in predicting mRNA transcription at multiple enhancer genes. By fitting additive, synergistic and logistic models to their time-course datasets, the authors find enhancers predicted to cooperate in an additive but also synergistic fashion, meaning that mRNA output of their assigned target gene(s) is higher than expected from the sum of eRNA transcription at cognate enhancers. Finally, they show that synergistic enhancers are mostly associated with cell-type specific genes and are enriched for cell-type specific TF binding motifs, although they do not show higher C/EBPα, chromatin accessibility or BRD4 levels, as previously reported for cell-type specific "super-enhancer" hubs.The authors address an important and timely question in the field of gene regulation, i.e. how do multiple enhancers cooperate to regulate their target genes. To achieve this goal, the authors produce a compelling and rich dataset, analyze various aspects of it and apply a modeling approach to assign enhancers to additive and synergistic modes.Overall, this study presents a comprehensive characterization of transcription and various chromatin aspects (chromatin accessibility, H3K4me1, factor binding) for a transdifferentiation timecourse from human B lineage cells to macrophages – a true tour-de-force. We however find that the authors partly overstate their findings and draw overly strong conclusions without adequately discussing caveats. We therefore recommend that the authors carefully revise their manuscript prior to publication to address two major concerns. The two major concerns relate to 1. some important untested assumptions that underlie the key finding of the manuscript, i.e. enhancer synergy and 2. partly overly strong conclusions that for example incorrectly imply causation from correlative analyses.

We thank the reviewer for the careful analysis and kind words. We carefully revised the manuscript to avoid overstating our findings and we now adequately discuss potential caveats.

1) Potential caveats to the key finding – enhancer synergyThe additive and synergistic cooperative behaviors reported by the authors heavily rely on (i) eRNAs – or TT-seq signals – accurately reflecting enhancer activity, (ii) a truly comprehensive set of enhancers, and iii) enhancer-promoter assignment strategies. The authors need to address or discuss these points in more depth, as their key finding, i.e. the existence of enhancer synergy, crucially relies on all three assumptions being correct – any underestimation of the number of enhancers or their activities will unavoidably make the enhancers appear synergistic even if they are not. We comment on each of these points separately and then discuss some implications in more depth below.(i) The authors treat this caveat rather superficially, referring to two recent publications (Henriques et al., 2018 and Mikhaylichenko et al., 2018) that study transcription at enhancers in *Drosophila*. Whether or not transcription rates truly reflect enhancer activities is far from clear: Mikhaylichenko et al. for example state "the levels and directionality of transcription are highly varied among active enhancers. […] enhancer RNA (eRNA) production and activity are not always strictly coupled", and in human HeLa and HepG2 cells, CAGE-eRNA-signals do not quantitatively correlate with enhancer activities (Andersson et al., Nature 2014; see supplementary figures 9a and 10a). In addition (more minor), at the core of TT-Seq is the transient sparse labeling of newly synthetized RNAs and their subsequent enrichment by affinity purification. We recommend discussing in more detail TT-seq's sensitivity and ability to reliably detect short, unstable and lowly abundant eRNAs (compared to more stable and longer mRNA molecules).

As recommend, we are now discussing in more detail TT-seq's sensitivity and ability to reliably detect short, unstable and lowly abundant eRNAs (compared to more stable and longer mRNA molecules. We also point out better the advantages of our approach.

Briefly, the generally used method to measure enhancer activity is to assess gene expression as a readout upon alterations of enhancers, for instance reporter gene assays or CRISPR editing of enhancers. Yet, the limitation of these methods is that we cannot easily quantify the enhancer activities and we cannot do the analysis genome-wide. We therefore used TT-seq to get genome-wide quantitative readouts for enhancer transcription, which is a good proxy for enhancer activity. TT-seq is highly sensitive to detect newly synthesized RNAs including unstable short RNAs such as eRNAs without perturbation (Schwalb et al., 2016). During T-cell stimulation, transcription from enhancers and promoters of responsive genes is activated simultaneously (Michel et al., 2017). Enhancers can be paired with their putative target gene promoters based on their proximity (Michel et al., 2017). In addition, we have shown that eRNA synthesis measured with TT-seq highly correlated with the putative target gene synthesis (Michel et al., 2017), implicating that enhancer transcription activity can indeed reflect enhancer activities. As eRNA production is a very good proxy for enhancer transactivation activity (Henriques et al., 2018; Mikhaylichenko et al., 2018), TTseq can be used to identify active enhancers, to pair enhancers with their putative target promoters, and to measure the transcription activity of enhancers and promoters genome wide.

ii) The authors define enhancers as TT-Seq Transcription Units (TUs) and further require them to be enriched for both H3K4me1 and ATAC-Seq signals. Through these latter criteria, only 8,165 of 26,130 TUs are defined as enhancers, leaving the vast majority of about 18,000 TUs unassigned. While stringent filtering makes much sense in general, a systematic underestimation of the number of enhancers might strongly confound the analysis of enhancer cooperativity. As the H3K4me1 mark is not required for enhancer activity (Dorighi et al., Mol Cell 2017; Rickels et al., Nature Genetics, 2017), the authors should consider whether these unannotated TUs might also stem from enhancer elements and e.g. from gene loci that show synergistic effects.

As suggested, we investigated whether unannotated TUs might also stem from enhancer elements. Among non-coding TUs, 13,952 TUs overlap with H3K4me1 signals, whereas 8,529 TUs overlap with ATAC-seq signals. The restriction rather comes from ATAC-seq, which is required for eRNA synthesis (Figure 2—figure supplement 1A). However, to exclude the possibility that the unassigned TUs are left out due to the low sensitivity of chromatin accessibility detection etc, we have paired these unannotated TUs with the nearest genes, and repeated the analysis to identify synergistic enhancers. After excluding ncRNAs that are downstream RNAs, convergent RNAs, upstream antisense RNAs or that overlap with more than 20% of transcripts annotated in Gencode, 16360 TUs (or 15487 extended EUs after merging TUs within 1kb) remained for further analysis (see Materials and methods for details). Using the neighboring pairing method, 1184 of 1790 TUs were paired with more enhancers than previous analysis. In total, we observed 544 and 209 genes were regulated by additive and synergistic enhancers, respectively. Of 136 synergistically regulated genes that have been previously identified, 71% (97 genes) were regulated in synergistic manner (Figure 5—figure supplement 1H). This indicates that the widespread synergistic behavior we observed was not due to an underestimation of the number of enhancers. We edited the text accordingly and this clarifies the reviewer’s concern.

iii) In addition to potentially missing enhancers by an overly stringent enhancer definition (point ii), the number of enhancers per gene can also be underestimated at the step of enhancer-to-gene assignment. To mitigate this problem, the authors use two complementary assignment strategies, "neighboring" and "1Mb", which make sense yet have their caveats. While the 1Mb method should be more inclusive, it for example imposes a 0.4 correlation coefficient between eRNA and mRNA changes, thus limiting the pertinence of this approach for further hypothesis testing (see major concern 2). More importantly though, the two approaches don't seem to yield very consistent results: 136 genes (92+44; subsection “Enhancer cooperation can be additive or synergistic”) were classified as synergistic by the neighboring method and 194 by the 1Mb method (subsection “Enhancer synergy is a robust phenomenon”), yet only 57 could be tested by both methods of which 33 were consistently classified as synergistic. While the authors claim that this corresponds to a strong overlap (a Fisher exact p-value is provided with 3.5e-11), it is unclear why so few genes can be tested by both methods and why of the 136 or 194 synergistically regulated genes, respectively, only 33 are consistent between methods. At the very least, these numbers suggest that the classification to the additive or synergistic models strongly depends on the enhancer-to-gene assignment.

The reviewer is correct and the relatively low number of consistent E-P pairs depends on the type of assignment. Briefly, to test enhancer synergy rigorously, we can only use genes that are paired with more than two enhancers. Since the number of enhancers paired to each gene vary between the neighboring and 1Mb methods, only a small number of genes could be tested for enhancer synergy. Nevertheless, our results are consistent between two methods. As in the response to the reviewer #2, we also employed a third pairing method by pairing enhancers to the correlating promoters within the same TAD obtained from Hi-C data. The synergistically regulated genes found with the neighboring and TAD pairing methods significantly overlapped, supporting our conclusions on synergistic enhancer action.This information was added to the text and figures.

With these caveats in mind, it is not clear whether the manuscript conclusively demonstrates the existence of synergistic enhancer cooperativity and what to recommend. A systematical under-estimation of enhancer activities or their dynamics (Figure 3B indicates that eRNAs are only detected rather late) would bias the analysis towards seemingly synergistic enhancer cooperativity, especially for genes that are highly expressed and/or strongly upregulated.

Although it is true that Figure 3B shows eRNAs are strongly synthesized at 96h in the LATE Cluster, eRNA synthesis is also observed at earlier time points in other clusters. In the originally submitted manuscript we had already excluded that there is a bias towards highly expressed genes by showing that synergistically regulated genes do not have systematically higher expression levels (Figure 5—figure supplement 2A), and that not all LATE upregulated genes are synergistically regulated. In summary, with all these controls, we conclusively demonstrate enhancer synergy at selected loci.

To reach strong conclusions, one would probably have to directly test combinations of enhancers or perform CRISPR-Cas9-mediated deletion of additive and synergistic enhancers with similar individual activities (as has been done in papers claiming or refuting enhancer synergy of e.g. superenhancers). One would expect that the deletion of an additive enhancer would have a limited impacted on downstream mRNA transcription and transdifferentiation efficiency, while the deletion of a synergistic enhancer would lead to stronger effects and might potentially impede transdifferentiation.

Please compare our answers to reviewer #1. We spent one year trying both the enhancer knockout approach and reporter gene assays, without success, due to the complex nature of the system, which combines transdifferentiation and enhancer cooperativity. We now resubmit a revised version that is substantially edited and hope the reviewer is fine with publication of this strongly revised version.

However, especially given the data-richness of the manuscript, such major additional efforts seem to be beyond the scope of this project. Maybe the authors could more openly discuss the caveats listed above and how they might confound the analyses and/or perform additional sanity checks, maybe some of the following:– simulations to assess the potential influence of underestimating enhancer activities or enhancer number on the additivity vs. synergy estimates. For example, the authors could determine how a simulated increase of eRNA transcription (e.g. by 5%, 10%, 20% or 50%) impacts the assignment of gene loci to additive, synergistic and/or logistic classes.

We thank the reviewer for stating that major additional efforts are beyond the scope of this project. We have carried out the suggested test. In the original manuscript we had included more enhancers around a few genes and showed that this did not change conclusions with respect to enhancer synergy. We have now done such testing more systematically and added a statement to the text. Briefly, we simulated increase of eRNA transcription and determined synergistic genes. Synergistically regulated genes were hardly affected by the increase of eRNA transcription (Figure 5—figure supplement 1G).

– a systematic analysis of potential correlations of synergistic genes with the number of enhancers. The author used gene loci containing 2 to 20 enhancers for model fitting and it would be informative whether synergistic loci overall contain more enhancers compared to additive ones.

We carried out this test and found that there is no difference in enhancer numbers between additive and synergistic loci (median 3.5 vs 3 enhancers, respectively) (ECDF plot). We added this to the manuscript (Figure 5—figure supplement 1H).

2) Data presentation and overly strong conclusionsThe authors describe and analyze their datasets by a combination of rule-based filtering and direct comparisons. This strategy is valid, yet can easily suffer from circular logic if the rules influence the comparisons, which can render results trivial. We recommend that the authors carefully revise their manuscript to remove analyses that might suffer from circular logic or discuss such caveats and tone down their conclusions.

We understand the concern and went through the manuscript and made sure there are not analyses that might suffer from circular logic.

For example, the authors require "nucleosome depletion" to define enhancers and subsequently study 4550 C/EBPa binding sites "with low or undetectable chromatin accessibility at 0h", which unavoidably leads to the observation that chromatin opens.

This may be a misunderstanding. We define enhancers based on enhancer transcription, which can only occur after nucleosome depletion because the Pol II machinery needs to access DNA. Many of these enhancers showed changes in activities (Figure 2C). Our question was how enhancers are newly activated from closed chromatin to active transcription. The basic premise here is that enhancers undergo chromatin opening. Then the question is on the temporal relationship of transcription factor binding, H3K4monomethylation, chromatin accessibility and transcription activities at the sites undergoing chromatin opening. In fact, this cannot be observed if chromatin does not open.

Similarly, the "1Mb" method to assign enhancers to promoters includes a requirement that the activities need to be correlated, which means that correlations (e.g. those shown in Figure 4—figure supplement 1 E-F) cannot be seen as results.

We agree and thank the reviewer for spotting this. We removed these figure panels and the corresponding text.

Similarly, the authors draw strong causal conclusions from correlative analyses (e.g. "Chromatin opening enables enhancer RNA synthesis”, or "Enhancer synergy can drive cell fate determination", title), which should be avoided.

We changed the title and trust the reviewer agrees to it. We edited the text to make sure there are no overly strong conclusions.

[Editors’ note: what follows is the authors’ response to the second round of review.]

Revisions:1) As the authors could not directly validate enhancer synergy, they indirectly infer its existence by the following logic: They show that the sum of enhancer activities does not match the transcription levels for some genes but leaves some gap. They use this gap to argue that the identified enhancers must be super-additive, i.e. synergistic, in order to fill this gap. While plausible, such an inference is different from the unambiguous demonstration of enhancer synergy. This fact will need to be better reflected in the Title and Abstract. Otherwise, a superficial reader might assume that synergy has been demonstrated directly. This wording could for example directly state the authors' reasoning, namely that a gap in observable enhancer activities suggests the existence of enhancer synergy for some genes. The last sentence of the revised Abstract needs to provide clarity on the comparison being made between synergistically functioning enhancer clusters and super-enhancers. It could read: "Enhancer synergy appears to be dependent on binding of cell type-specific transcription factors and such interacting enhancers are not predicted from occupancy or accessibility data that are used to detect super-enhancers."

We appreciate the comment and changed the Abstract as suggested. We also changed the title to: “Evidence for additive and synergistic action of mammalian enhancers during cell fate determination”.

2) The claim that enhancer synergy "could not be predicted from occupancy or accessibility data that are used to detect super-enhancers" hinges on the comparison of eRNA-derived enhancer cohorts with super-enhancers. This comparison retains two issues that still need to be addressed. First, super-enhancers represent regions of high enhancer activity, originally defined by Med1 or H3K27ac ChIP-seq. Consequently, open chromatin (ATAC-seq) or the locations of a pioneer factor (C/EBPa) that induces open and poised – but not necessarily active – chromatin is not necessarily expected to yield meaningful results in terms of defining genomic regions with cell type-specific and lineage-defining enhancer and gene activity. This would leave the BRD4 ChIP-seq data as the sole signal to define meaningful super-enhancers. However, its high peak count of >98,000 raises concerns about potential ChIP quality issues that unfortunately cannot be assessed because the authors fail to provide additional information on these ChIPs other than the peak count. That the Spearman rho values for BRD4 ChIP replicates fluctuate between 0.72 and 0.99 also do not inspire confidence. Here, a H3K27ac as a broader and more robust ChIP-seq marker of enhancer and transcription activity might help to improve super-enhancer calling, should BRD4 turn out to be sub-optimal. Second, save for a handful of super-enhancer-associated genes that have been included in the revised manuscript, the biological relevance (i.e. gene ontology term enrichment) of the identified super-enhancer-associated genes cannot be verified by readers, since the authors provide neither the corresponding list of genes nor an analysis of the super-enhancer-enriched GO terms. At a minimum, the authors should provide these gene lists for the different super-enhancer methods and additional quality information of the ChIP-seqs used to identify super-enhancers, perform GO term analyses and provide the full list of significantly enriched GO terms, to be able to gauge whether they identify true super-enhancers with the corresponding expected characteristics (cell type specificity, enrichment near lineage-determining genes and transcription factors), and whether these are indeed distinct from synergistic enhancers.

To provide more support that classically defined superenhancers do not correspond to the synergistic enhancers, we used H3K27Ac ChIP-seq data (Stik et al., 2020) as suggested by the reviewers. Consistent with our previous results, the superenhancers derived from H3K27Ac ChIP-seq signal included both synergistic and additive enhancers (Figure 5F, Figure 5—figure supplement 4A). Similar to superenhancers derived from other markers for high enhancer/transcription activity, RNA synthesis from the superenhancers which show high H3K27Ac levels at 96h and their paired promoters was not always the highest at 96h compared to other time points (Figure 5—figure supplement 4C). The H3K27Ac level at synergistic enhancers was similar to the level at additive enhancers (Figure 5G).

Furthermore, we included a list of genes regulated by superenhancers derived from various ChIP-seq data in Supplementary file 1. To ensure these superenhancers correspond to the previously reported superenhancers, we compared the list to dbSUPER (Khan and Zhang, 2016). Most of the superenhancers identified in this study could be found in the database (Supplementary file 1). Additionally, GO analysis was performed using superenhancers. GO terms with p values less than 0.05 were included for superenhancers derived from eRNA signal, ATAC-seq signal, BRD4 and C/EBPa ChIP-seq signal in Supplementary files 2, 4 and 5. GO terms with Bonferroni p value less than 0.05 were included for superenhancers derived from H3K27Ac ChIP-seq signal in Supplementary file 3. The resulting GO terms included immune response, inflammatory response and defense response. Finally, FRiP of ChIP-seqs can be found in Supplementary file 7. Additional genome browser screenshots are included in Figure 4—figure supplement 1E and Figure 5—figure supplement 4B.

3) The analysis of transcription factors in synergistic enhancers versus additive ones is not fully developed and contains inaccuracies and confusing interpretations of the data. Specifically, there is confusion over the two motifs that were determined to be enriched in additive enhancers, for FOS and E2F family factors.a) The motif for "FOS family" factors (TGAc/gTCA) is actually the motif to which most AP-1 family heterodimers (Fonseca, Nat Comm 2019). Indeed, AP-1 family factors (and not just FOS) play cell type-specific roles in macrophage development and function and their activity is low or undetectable at B cell enhancers by motif analysis (Heinz et al., Mol Cell 2010). Consequently, the statement that "additive enhancers were not [enriched for macrophage-specific TFs]" is incorrect, especially in the context of this study of B cell (-like cell) trans-differentiation into macrophage-like cells, and it is not questionable whether FOSL1 as given in Figure 6—figure supplement 1C is indeed the factor that binds these motifs, and which of the Jun-like factors that become upregulated are involved in AP-1 heterodimer binding to them.

The reviewers are right that the motif for FOS family factors is bound by most AP-1 family heterodimers, which have cell type specific roles in macrophage development. We had mentioned this in the originally submitted manuscript. However, this motif was enriched in additive enhancers only at 12h when the cells did not establish macrophage specific program (Figure 6—figure supplement 1B). At later time points in macrophage-like cells, these motifs were found both in additive and synergistic enhancers as no enrichment was shown in Figure 6—figure supplement 1B. We interpret this to mean that the TFs in FOS family and AP-1 family bind both in additive and synergistic enhancers. We improved the text to avoid any misunderstanding.

b) There appears to be a disconnect between the interpretation of the authors' finding that E2F motifs are enriched in additive enhancers, their statement that E2F factors have regulatory functions in macrophages and the expression data in Figure 6B showing that E2F factors are rapidly downregulated upon estrogen addition, which is in line with the cell cycle arrest observed in these cells when leaving the B cell-like state (Rapino, 2013). Together, this indicates that E2F factors would only be able to bind to additive enhancers and play a role in their activation in the proliferative B cell-like state, but not in the macrophage-like state. If so, then why are E2F motifs enriched in late-time point enhancers?

E2F factors have general regulatory functions in many cell types including macrophages. Indeed, E2F1, a member of E2F factors, is rapidly downregulated upon estrogen addition. Yet, E2F3, E2F4 and E2F6, other members of E2F factors, maintains their expression level during transdifferentiation. Thus, it is possible that E2F motifs were enriched in late-time point enhancers due to the E2F factors without significant expression changes in macrophage-like cells such as E2F3, E2F4 and E2F6.

4) The number of C/EBPa ChIP-seq peaks is surprisingly low for an overexpressed pioneer transcription factor, especially while other ChIPs in the data set have unusually large numbers of peaks (>98,000 peaks for BRD4 seems very high for a non-DNA-binding coregulator). These peak numbers are unusual and raise data quality concerns, and for C/EBPa might also explain the small number of C/EBPa-defined "super-enhancers". Further, the peak numbers identified by ATAC-seq are not provided, and here the Spearman's rho was only calculated for promoters, which is problematic in a manuscript dedicated to enhancers and raises additional data QC concerns. Providing additional QC data such as FRiP and missing peak numbers, preferentially in a tabulated form, as well as additional genome browser track screenshots at different resolutions (also of the BRD4 ChIP-seqs) are recommended.

As additional QC data, peak numbers for ATAC-seq, Spearman’s rho for all ATAC-seq peaks and FRiP of ChIP-seq data are provided in Supplementary file 7. Genome browser screenshots of the datasets including BRD4 and H3K27Ac around CD14 are shown in Figure 4D and Figure 4—figure supplement 1). Genome browser screenshots around IRF8 and EBF1 are also shown in Figure 5—figure supplement 4B (Also mentioned in the additional revision point).